# Molecular characterization of a marine turtle tumor epizootic, profiling external, internal and postsurgical regrowth tumors

Kelsey Yetsko[1,12], Jessica A. Farrell [1,2,12], Nicholas B. Blackburn [3,4,11], Liam Whitmore[1,5], Maximilian R. Stammnitz [6], Jenny Whilde[1], Catherine B. Eastman[1], Devon Rollinson Ramia[1], Rachel Thomas[1], Aleksandar Krstic[7], Paul Linser[1], Simon Creer [8], Gary Carvalho[8], Mariana A. Devlin[9], Nina Nahvi[9], Ana Cristina Leandro[3,4], Thomas W. deMaar[10], Brooke Burkhalter[1], Elizabeth P. Murchison[6], Christine Schnitzler [1,2] & David J. Duffy [1,2,5,7,8 ✉]

Sea turtle populations are under threat from an epizootic tumor disease (animal epidemic) known as fibropapillomatosis. Fibropapillomatosis continues to spread geographically, with prevalence of the disease also growing at many longer-affected sites globally. However, we do not yet understand the precise environmental, mutational and viral events driving fibropapillomatosis tumor formation and progression.

Here we perform transcriptomic and immunohistochemical profiling of five fibropapillomatosis tumor types: external new, established and postsurgical regrowth tumors, and internal lung and kidney tumors. We reveal that internal tumors are molecularly distinct from the more common external tumors. However, they have a small number of conserved potentially therapeutically targetable molecular vulnerabilities in common, such as the MAPK, Wnt, TGFβ and TNF oncogenic signaling pathways. These conserved oncogenic drivers recapitulate remarkably well the core pan-cancer drivers responsible for human cancers. Fibropapillomatosis has been considered benign, but metastatic-related transcriptional signatures are strongly activated in kidney and established external tumors. Tumors in turtles with poor outcomes (died/euthanized) have genes associated with apoptosis and immune function suppressed, with these genes providing putative predictive biomarkers.

Together, these results offer an improved understanding of fibropapillomatosis tumorigenesis and provide insights into the origins, inter-tumor relationships, and therapeutic treatment for this wildlife epizootic.

_______________

A list of author affiliations appears at the end of the paper.

Sea turtle fibropapillomatosis (FP) is potentially a canary in the coalmine, indicating that continued human-induced environmental damage may be an alternative route by which oncogenicity is conferred on normally well-tolerated viruses. This is particularly worrying as long-lived reptiles usually have robust anti-cancer defenses[1,2], and as there is already a range of human viruses known to be capable of inducing tumor formation when the host immune system is compromised[3,4]. The application of precision oncology to FPcan reveal the precise mechanisms through which environmental triggers, viral dynamics, and host cell transformation can rapidly induce novel cancer incidence on an epidemic scale, thereby simultaneously informing human and wildlife cancer research[2,5]. Precision oncology incorporates recent advances in -omic technologies (genomics, transcriptomics, proteomics, metabolomics, high-throughput histology/imaging etc.) and computational advancements and applies them to the molecular profiling of tumors to provide mechanistic clarity, to identify targetable alterations and predictive biomarkers, and to direct the correct treatments to responsive patient cohorts[6–8]. Precision oncology is rapidly developing and has entered the mainstream of human clinical practice[6–8].

FP (Fig. 1a) is a tumor disease of epizootic (animal epidemic) proportions, affecting wild populations of endangered sea turtles circumglobally[9–11]. Sea turtle FP continues to spread geographically throughout equatorial and subequatorial regions, and has now been reported in every major ocean basin in which green turtles (*Chelonia mydas*) are found, particularly in nearshore habitats (www.cabi.org/isc/datasheet/82638)[10,12–17]. In addition to spreading globally, FP rates continue to increase in many affected sites, posing serious conservation challenges. Reportedly, of all green sea turtles stranding, over 40% in Florida, 30% in Hawaii, 35% in Texas, 35% in northeastern Brazil, 34% at Príncipe Island in the Gulf of Guinea, and 50% in Puerto Rico are FP-afflicted[12,18–22]. Many of these sites have seen a rapid increase in disease prevalence over recent years; for instance, from 13.3 to 42% (2005–2016) in Florida, 13.2 to 35.3% (2012–2015) in northeastern Brazil, and 0 to 35.2% (2009–2018) in Texas, with the occurrence of FP in Texas beginning in 2010 at a rate of 0.6%[12,18–21]. In contrast, the incidence of FP in Hawaii has been declining, as of 2014 (the most recent year on record) 44% of stranded turtles had FP[12,23]. The declining prevalence in Hawaii has been postulated to be due to a culling strategy or to biologically distinct viral strains with altered shedding dynamics[18,23,24].

A chelonian-specific herpesvirus (chelonid alphaherpesvirus 5 (ChHV5)), has been implicated in driving the FP disease epizootic, although Koch's postulates to confirm its causative role have yet to be definitively confirmed[10,25–28]. Even so, ChHV5 infection alone is not sufficient to induce FP tumor growth[29–33]. An anthropogenic-linked environmental co-trigger(s) may be the required key to both the development of FP tumors and the geographic spread of the disease[10,28,32,34]. Much uncertainty remains about the postulated environmental trigger(s) and how they interact with ChHV5 and the host immune system to give rise to FP tumorigenesis[10]. There is a paucity of knowledge concerning the molecular signaling events underpinning FP tumor initiation, development and growth, with even less known about the viral and host transcriptional landscape driving tumorigenesis. In addition, the precise relationship between visceral internal tumors and the more common external tumors remains to be elucidated. Analysis of the viral aspects of the current study's transcriptomic and genomic data are explored in a companion paper on ChHV5[35].

While advances in our understanding of the FP tumor disease epizootic in sea turtles continue to be made[10,28,33,36–43], many questions remain unanswered in relation to this enigmatic disease. There is virtually no molecular information about the relationship (e.g., primary/metastatic) between the numerous tumors, which can range from tens to hundreds, arising on a single individual turtle. Similarly, the molecular drivers of early-stage, internal, and postsurgical regrowth tumors remain to be elucidated. Determining the contribution of each facet of this multifactorial disease will be key to combatting this epizootic disease both at the level of individual clinical treatment and for population-level management/mitigation strategies[2,33]. The relative contributions of ChHV5, environmental trigger(s), immune suppression, host genome mutation and altered gene expression to FP oncogenesis have never been determined. Here, we applied a combination of extensive transcriptomics, precision oncology[7,33,44], and immunohistochemistry to determine the host molecular events underpinning FP tumorigenesis, growth, and postsurgical recurrence. In particular we focused on the profiling and contrasting of the host transcriptional events responsible for driving five types of FP tumor: new external, established external, postsurgical regrowth external, internal visceral lung, and internal kidney tumors.

## Results

### Divergent transcriptomes of internal and external fibropapillomatosis tumors

To determine the molecular events governing FP tumor growth, and whether different oncogenic signaling networks drive tumor types in different tissues, we conducted transcriptomics of FP tumors and patient-matched nontumor tissue (90 RNA-seq samples, Supplementary Data 1). Differentially expressed (DE) genes were then analyzed at the gene, pathway, network, and systems levels. When the mRNA transcripts DE between external new growth, external established, and external postsurgical regrowth (recurring) FP tumors were compared, there was a high degree of overlap (Fig. 1b). The overlap suggests that the molecular events driving external FP tumor formation, growth, and regrowth are broadly similar.

Next we compared the DE transcripts between established external tumors and those of internal visceral tumors, lung FP and kidney FP (Fig. 1c, Supplementary Data 2). In contrast to the three types of external tumors (new growth, established, regrowth, Fig. 1b), the internal visceral tumor DE transcripts were extremely divergent to the external tumor DE transcripts (Fig. 1c). Furthermore, lung and kidney tumor transcriptomes had minimal overlap with each other (Fig. 1c) suggesting that different oncogenic signals are driving internal FP compared with external FP tumors, as well as kidney FP tumors compared with lung FP tumors. The diverging transcriptional profiles of internal and external tumors is also apparent at the whole transcriptome level (Fig. 1d, e).

A large number of genes were DE between all tumor samples and all nontumor samples (Fig. 2a). Highly upregulated genes included those associated with oncogenic signaling, such as *Cthrc1* (Wnt and adenocarcinoma signaling) and *Crabp2* (retinoic acid (RA) signaling and embryonal carcinoma-associated gene), while highly downregulated genes included those associated with skeletal muscle (*Aacta1*, *Myl6b*, and *Klhl41*, Fig. 2a). *CRABP2* (Fig. 2a) is a predictive biomarker for a number of human tumors, such as ovarian cancer and non-small cell lung cancer[45,46]. Therefore, we examined *Crabp2* expression across our turtle patient cohort. Patients with positive case outcomes (release) tended to have tumors with lower *Crabp2* expression when compared with patients that died in care or were

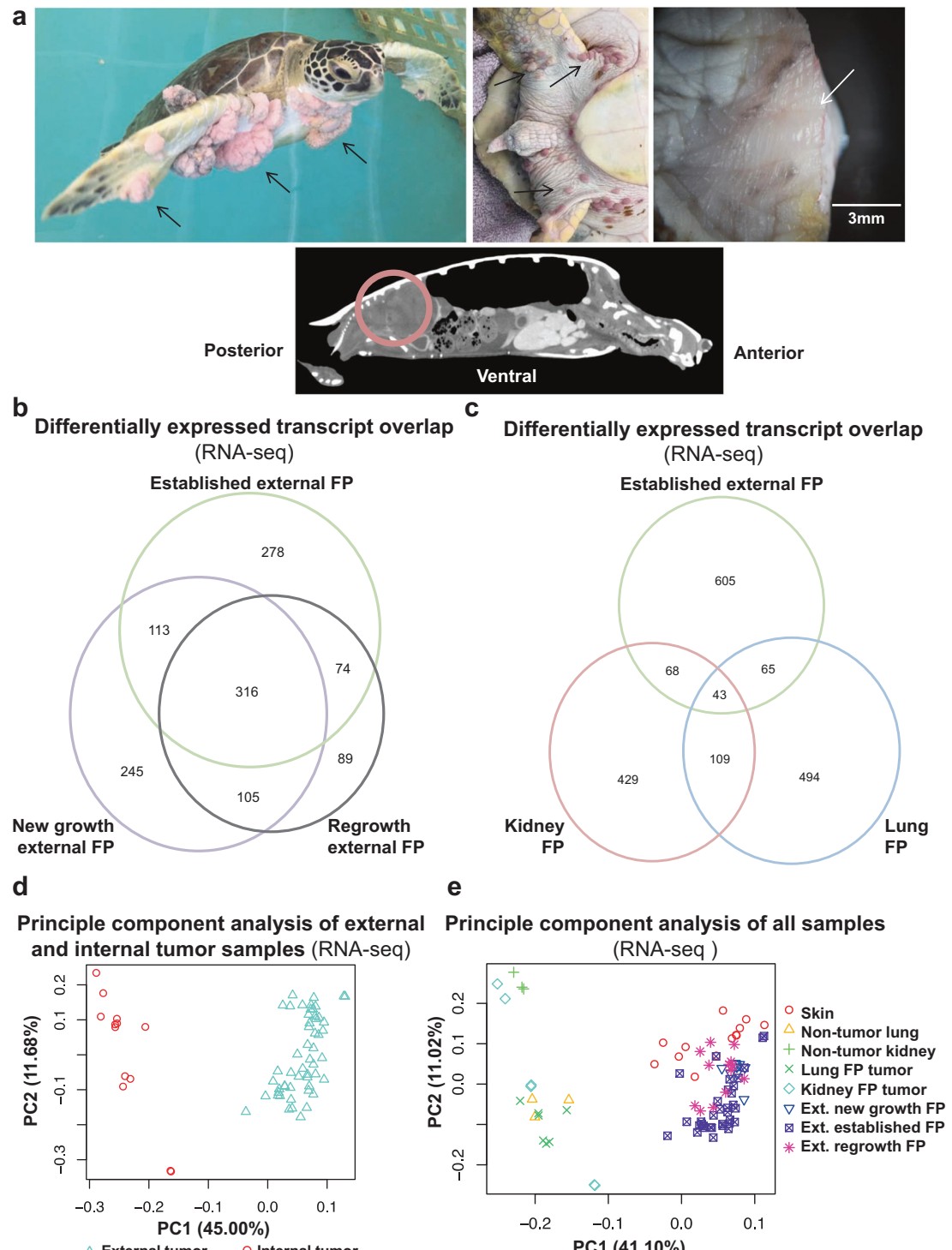

**Fig. 1 Fibropapillomatosis tumors and differential transcript expression. a** (Top, left) Fibropapillomatosis-afflicted green sea turtles (*Chelonia mydas*) in one of the hospital's seawater tanks, awaiting tumor removal surgery. Established tumors are visible as large pinkish outgrowths. Top, middle: Numerous new growth tumors occurring around the ventral tail and rear flipper area of patient 25-2018-Cm 'Lilac'. Top, right: Postsurgical regrowth tumor imaged after surgical resection. Regrowing tumor is the pinkish tissue, and is surrounded by paler non-tumored skin. Bottom: A computed tomography (CT) scan of fibropapillomatosis-afflicted *C. mydas*. CT is one of the approaches used for diagnosing internal tumors. A large kidney tumor is circled. In all other images, arrows indicate selected examples of external tumors. **b, c** Overlap of transcripts significantly differentially expressed (DE) (as called by DESeq2) in fibropapillomatosis from the RNA-seq data. Transcripts were considered significant if passing the following cutoffs: adjusted *P* value of <0.05 and $\log_2$ fold change of >2 or ≤2. **b** Overlap of DEs from the following comparisons: established external FP, new growth external FP, and regrowth external FP, when all are compared to healthy skin for differential expression analysis. **c** Overlap of DEs from the following comparisons: established external FP, kidney FP, and lung FP, each compared to their non-tumored tissue sources for differential expression analysis (healthy skin, healthy kidney, and healthy lung, respectively). **d** Principle component analysis (PCA) of all internal tumor samples compared to external tumor samples, RNA-seq. **e** PCA of all samples, includes all tumor and nontumor samples, RNA-seq. In the figure key external is abbreviated as 'Ext.'.

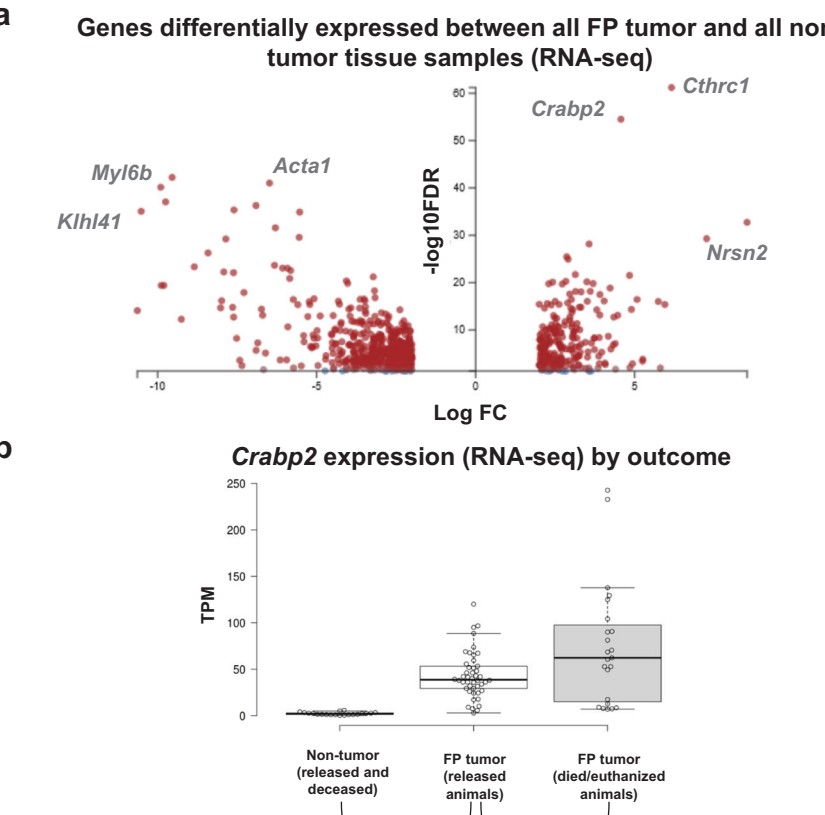

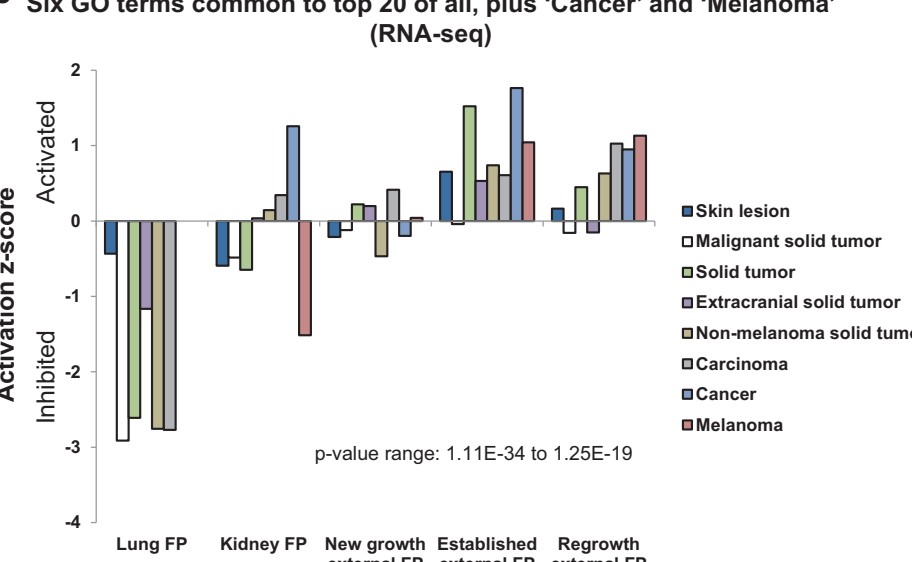

**Fig. 2 Gene level and gene ontology (GO) term analyses of differentially expressed transcripts in each tumor type. a** Volcano plot of genes differentially expressed between fibropapillomatosis tumors (all types) and nontumor tissue (all types) as determined by DESeq2 analysis of the RNA-seq samples. Transcripts were considered significant if passing the following cutoffs: adjusted $P$ value of <0.05 and $\log_2$ fold change of >2 and ≤2. **b** Expression levels of CRABP2, as detected by RNA-seq, in nontumor tissue and tumors of patients with varying rehabilitation outcomes. $N = 89$ samples. Per outcome: released = seven turtles; died/euthanized = five turtles. Box plot with Tukey whiskers. $P$ values of inter-group comparisons are shown below the x-axis (Mann–Whitney $U$ tests). $n = 89$: 20 nontumor, 23 poor-outcome tumor and 46 good outcome tumor samples. **c** Activation/inhibition $z$ scores of the six GO terms common to the top 20 GO terms of all sample comparisons, plus the 'Cancer' and 'Melanoma' GO terms, as detected by IPA, ranked by $P$ value (calculated by right-tailed Fisher's exact test, with Benjamini–Hochberg correction).

euthanized due to advanced disease (Fig. 2b), though for this cohort this difference fell just short of statistical significance (Mann–Whitney $U$ test, $P = 0.0518$).

**Strong association between sea turtle tumor transcriptomes and cancer gene ontology terms.** To better understand the main biological processes involved in each tumor type we examined the top 20 Gene Ontology (GO) terms (called by Ingenuity Pathway Analysis (IPA), ranked by $P$ value) for each set of DE transcripts (Supplementary Fig. 1a–e). Eighteen of the top 20 GO terms associated with established external FP tumors were cancer or neoplasia-associated terms, with 'activation of skin tumors' and 'melanoma' associated terms featuring prominently (Supplementary Fig. 1a). The remaining two noncancer GO terms were related to muscle, and were inhibited. This finding helps to validate the analysis pipeline (GO term results were called without the input of any a priori FP knowledge) as FP tumors are known to contain less muscle tissue than non-tumored skin punch biopsies which also contain subcutaneous tissue[47–49]. For analysis of the top 20 GO terms for the other tumor types, new growth external, regrowth external, kidney FP and lung FP, see Supplementary Figs. 1b–e and 2a–c. Overall, of the top 20 GO terms from each tumor compared with their matched nontumor tissue type, six were common to every FP type sequenced, although their activation/inhibition status differed between internal and external tumors (Fig. 2c). These shared top GO terms were all cancer-associated (Fig. 2c).

**Hosts attempt to mount an immune response to fibropapillomatosis.** To further determine what signaling events are prominent in driving early external tumor formation (new growth and postsurgical regrowth), we examined the top 200 (ranked by $P$ value) GO terms in early external tumors compared with established external tumors (Fig. 3a). Similar to the gene level analysis there was a very high degree of overlap between the three external tumor types. However, of the 200 GO terms, 22 were uniquely common to early new and regrowth tumors (Fig. 3a). Of these 22 GO terms, eight of them were associated with leukocyte/lymphatic processes (Fig. 3b), suggesting a crucial role of immune response during initiation of FP tumor growth. All eight leukocyte/lymphatic process GO terms were strongly activated across the three external FP types (Fig. 3b), although for established external tumors these terms fell outside of the top 200 GO terms. Interestingly, only two of these eight terms ('cell movement of leukocytes' and 'leukocyte migration') were called as statistically significant in internal tumors (although outside of the top 200, Fig. 3b). In kidney FP tumors, both terms were strongly activated, mirroring the external tumors. Conversely however, both of these GO terms were inhibited in lung FP tumors (Fig. 3b), again highlighting differences between the molecular signaling events driving internal and external FP tumors. Interferon gamma, a cytokine critical to both innate and adaptive immunity, was also called as an inferred transcriptional regulator (ITR) in all five tumor types, being activated in every tumor type barring lung tumors, where it was inhibited (Supplementary Fig. 2d)

Given the strong leukocyte/lymphocyte infiltration results from the transcriptomic profiling of early FP tumor types, we next assessed the host adaptive immune response by identifying CD3 positive T lymphocytes[50] within FP tumor tissue sections. The lymphocyte analysis confirmed the transcriptomics findings, revealing that in early-stage FP tumors there is a high level of T-lymphocyte infiltration (Fig. 3c). CD3 is an immunophenotypic cell marker, which is found only in T lymphocytes and is central

to the formation of antigen-receptor interactions through the T-cell receptor/CD3 complex[50,51]. CD3 positive staining was strongest in epidermal regions, where inclusion bodies (presumably due to lytic ChHV5) most commonly occur within FP tumors[25,52]. CD3 staining was strongest in new growth tumors (Fig. 3c), but weakened in more advanced tumors (Supplementary Fig. 3a). Together the transcriptomics and CD3 staining demonstrate that an early immune response is mounted by the host (*C. mydas*), either to the tumor cells themselves, and/or ChHV5 infection.

**Conserved internal and external tumor oncogenic signaling networks and therapeutic vulnerabilities.** The disparate signaling events detected by the transcriptomics between external, lung and kidney tumors potentially make it less likely that a single systemic anti-cancer therapeutic would prove effective against both external and internal tumors. However, to investigate whether any common therapeutically targetable oncogenic pathways exist between these tumor types, we next compared their top 100 ITRs. IPA analysis infers the upstream transcriptional regulators responsible for the observed transcriptomic signatures by comparing the differential gene expression profiles to known regulator induced changes in its knowledgebase. Mirroring the gene-level analysis, ITR analysis also showed very little overlap between the top 100 ITRs of established external, lung and kidney FP tumors (Fig. 4a). However, if a common therapeutically targetable vulnerability exists it should be located in the overlapping ITRs of these three FP tumor types. Therefore, we further investigated the 16 ITRs common to all three FP types (Fig. 4a–c). These 16 ITRs represented nine genes and seven pharmaceutical compounds (drugs). Of the nine gene ITRs, almost all were activated across all five FP tumor types sequenced, with new growth external tumors tending to be the exception (Fig. 4c). These nine genes form a highly interconnected regulatory network (Fig. 4b), with 32 edges between the nodes and a protein–protein interaction enrichment $P$ value of 3.82E-07 (STRING). Interestingly, RA signaling was activated strongly in established external, external regrowth, and kidney FP tumors (Fig. 4c). Retinoid therapy, to activate RA signaling, is widely used as an anti-cancer therapeutic and maintenance therapy for a number of human cancers (such as the pediatric cancer neuroblastoma[53]), although conversely, RA signaling is known to be activated in other cancer types. RA is a widely available and inexpensive anti-cancer therapeutic, which would make it ideally suited for use in sea turtle rehabilitation facilities. Unfortunately, the findings here suggest RA would be ineffective against FP, instead suggesting the converse, that RA signaling inhibition would be a possible target in FP. The transcriptomic findings explain the failure of early attempts to treat FP tumors and prevent regrowth using ectopic RA application (Supplementary Fig. 3b).

The 16 ITRs (Fig. 4c) tended to fall into three main categories: canonical Wnt signaling (Wnt3a and β-catenin), MAPK signaling (p38 MAPK, U0126, and SB203508), and immune-related signaling (CD44, IL6, APP, TNF, TGFβ, and dexamethasone). These pathways form part of an interlinked signaling network (Fig. 4b). We next examined β-catenin protein cellular localization within the tumors as a readout of pathway activity. As predicted by the transcriptomics, β-catenin was located at the cellular membrane in new growth external FP tumors, indicating inactivation of Wnt/β-catenin signaling (Fig. 4d). Again in line with the ITR analysis, nuclear localization of β-catenin was present within other tumor types (Fig. 4d and Supplementary Fig. 4a), which is indicative of Wnt/β-catenin pathway activation.

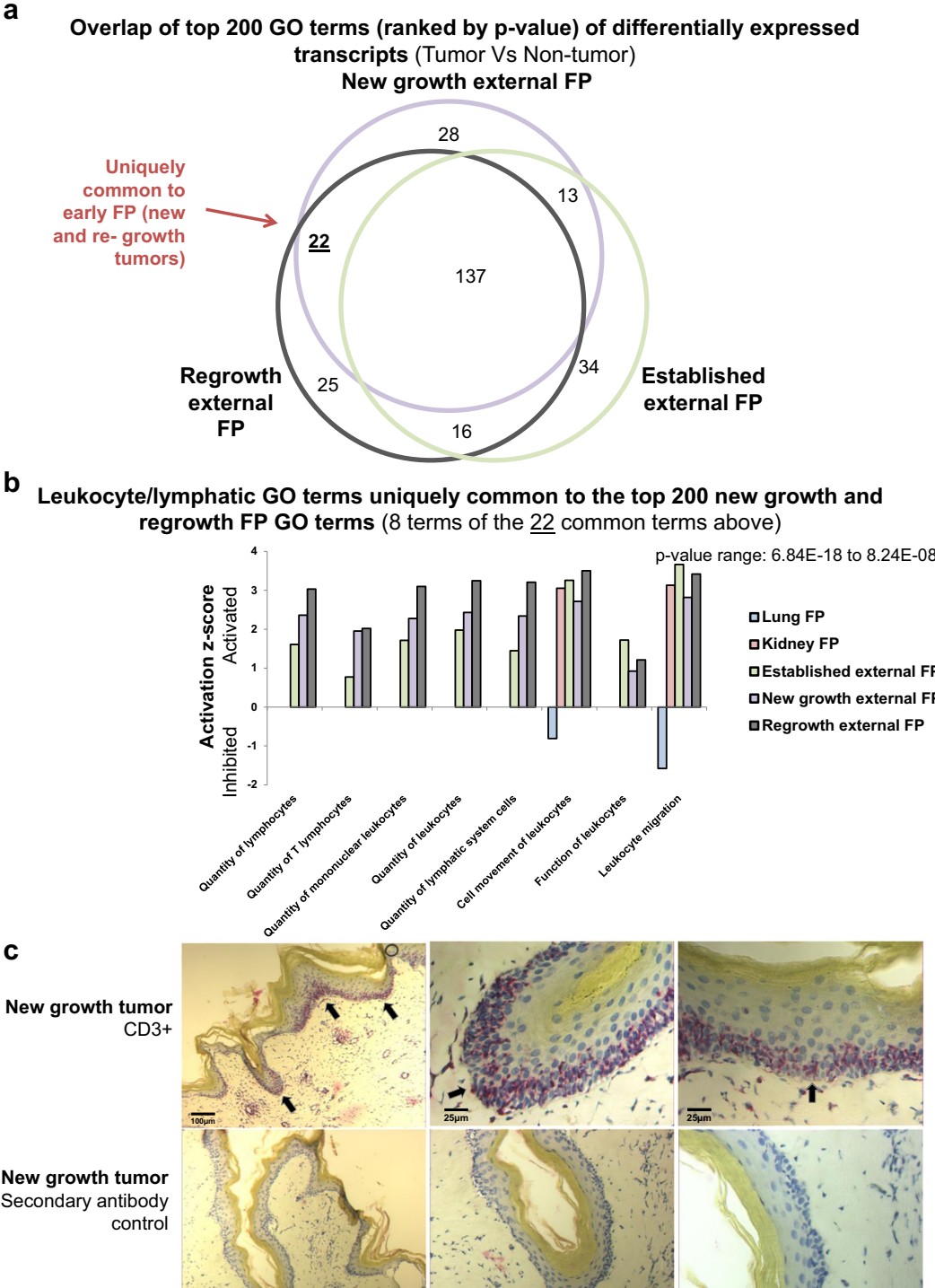

**Fig. 3 Transcriptomic- and histological-based immune profiling of fibropapillomatosis tumors. a** Overlap of the top 200 disease-associated GO terms associated with transcripts differentially expressed in different growth stages of external FP tumors (new growth, regrowth, established), as detected by IPA, ranked by *P* value (calculated by right-tailed Fisher's exact test, with Benjamini–Hochberg correction). Both activated and inhibited GO terms for each tumor type when it was compared to its healthy tissue source were included. **b** Activation/inhibition *z* scores for eight (the leukocyte/lymphatic-associated GO terms) out of the 22 disease GO terms uniquely common to the top 200 ranked GO terms of early fibropapillomatosis (new growth and regrowth, see **a**), shown for all tumor types. Some of these GO terms were called for lung, kidney and/or established external tumors (as shown), although they fell outside of the top 200 ranked GO terms called for these three tumor types. **c** CD3 antibody-based staining (red/purple) of T-lymphocyte infiltration in new growth tumor tissue, nuclei are counterstained with hematoxylin (blue staining). Selected positive CD3 stained areas are indicated by black arrows.

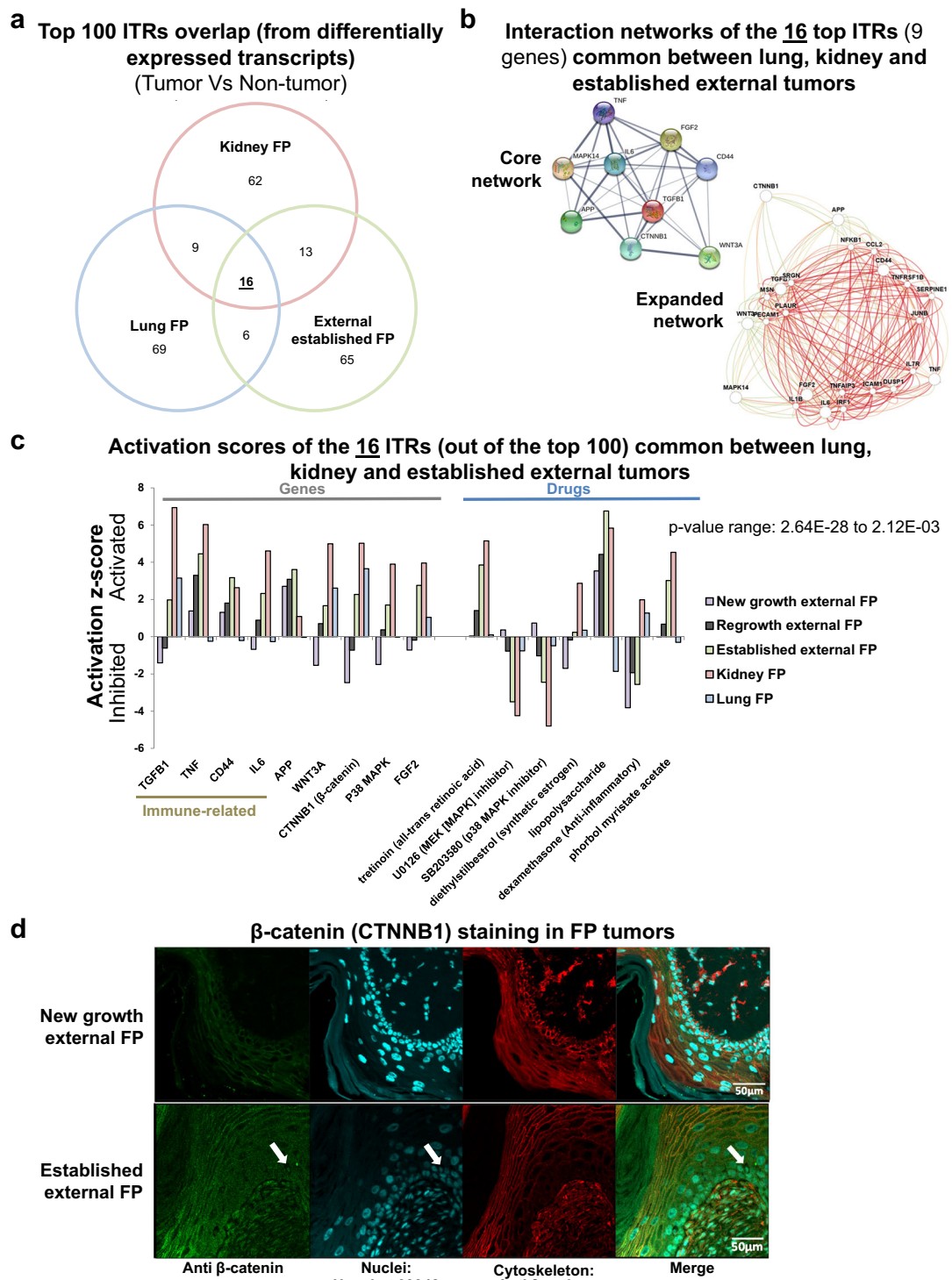

**Fig. 4 Transcriptional regulator analysis revealing the cellular signaling driving fibropapillomatosis and potential therapeutic targets. a** Overlap of the top 100 inferred transcriptional regulators (ITR) of the transcripts differentially expressed in different types of FP tumors (kidney FP, lung FP, external established FP) when compared to their respective non-tumored tissue sources, as detected by IPA, ranked by $P$ value (calculated by right-tailed Fisher's exact test, with Benjamini–Hochberg correction). The 16 ITRs that were common to lung, kidney, and established external FP were selected for further analysis. **b** Interaction networks of the top 16 ITRs (nine genes shown, seven drugs excluded) common between lung, kidney and established external tumors. Core network generated by STRING[94] (https://string-db.org/), expanded network generated by HumanBase (https://hb.flatironinstitute.org/). **c** Activation/inhibition $z$ scores for the 16 ITRs common between lung, kidney, and established external tumors, shown for all tumor types. ITRs are segregated according to functional class, i.e., genes and drugs. **d** New growth and established external tumor tissue sections, stained with anti β-catenin (an ITR called as activated for three of the five tumor types, see **c**) and counterstained with Hoechst 33342 to visualize nuclei and Anti β-actin to visualize cellular cytoskeletons. Selected cells with nuclear (activated) β-catenin staining are indicated by white arrows.

While nuclear β-catenin did occur in external established, regrowth, and internal tumors, it was far from ubiquitous, suggesting intra-tumor heterogeneity in terms of Wnt signaling activation. In external tumors, β-catenin was also strongly localized to the cell membrane in epidermal tumor cells (Supplementary Fig. 4b). Nuclear β-catenin was also detected in dividing nuclei of new growth tumors (Supplementary Fig. 4b). Generally the transcriptomics revealed that each of the three main shared signaling pathways (Wnt signaling, MAPK signaling and immune-related signaling) were activated; inversely, but logically, transcriptomic patterns of drug inhibitors of these pathways were inhibited (U0126, SB203580, and dexamethasone, Fig. 4c). This suggests that FP tumors (both external and internal) may be susceptible to treatment with inhibitors of these pathways.

**Transcriptionally-inferred molecular origins of fibropapillomatosis**. To gain further insights into FP's origins, we employed larger-scale network clustering analysis of the ITRs of the DE transcripts. Established tumors demonstrated clusters of 'viral and inflammatory responses', 'inhibition of anoikis' (programmed cell death of anchorage-dependent cells that detach from the extracellular matrix), 'cellular senescence', and 'miRNA regulation' (Fig. 5a). Interestingly, given FP's as-yet unidentified environmental trigger(s), the highly interconnected ITR network of established external tumors (protein–protein interaction enrichment $P$ value <1.0E−16) also had signatures related to 'cellular responses to organic substances and chemical stimulus' (Supplementary Fig. 5a, b). Furthermore, pathways and DE genes related to 'Kaposi sarcoma-associated herpesvirus infection' (KEGG pathway analysis FDR = 1.20E−32) were also detected in established external FP (Supplementary Fig. 5a). Kaposi sarcoma is a human herpesvirus-associated cancer which occurs in immunocompromised patients, for example HIV/AIDS patients[54]. Immune-related processes featured in all five FP tumor types (Fig. 5a, b and Supplementary Fig. 6a–c). Although transcriptionally divergent to established external FP tumors, lung FP also showed an interconnected network (protein–protein interaction enrichment $P$ value < 1.0E−16) with cellular immune response and organic substance and chemical response nodes (Supplementary Fig. 5b). Furthermore, clusters related to response to inorganic substances, metal ions, viruses, and radiation were also detected (Fig. 5b). Indeed, 'Quantity of Metal' was called as a GO term and was activated for all tumor types, with the exception of kidney tumors in which it was not called (Supplementary Fig. 6d).

Pathway analysis (Fig. 5c) and GO term analysis (Fig. 5d) revealed a graded activation of metastatic-related signaling across the FP tumor types. FP tumors have been described as primary, despite the numerous tumors which regularly develop on each afflicted individual. However, to date, no in-depth molecular analysis has been conducted to determine if all tumors on an individual are indeed primary, or if all or some of them (particularly internal visceral tumors) occur due to metastatic spread of a primary tumor. Our transcriptomics suggests that at a minimum, FP tumors have a propensity to mutate towards the activation of metastatic pathways, with kidney tumors showing stronger activation than external tumors (Fig. 5c, d). Early external FP tumors do not display metastatic signaling, rather such pathways are mildly inhibited (Fig. 5c, d). Established external FP show mild metastatic signaling activation, while internal tumors, particularly kidney tumors, show elevated activation of these pathways (Fig. 5c, d). This suggests that external tumors may acquire metastatic propensity over time. It should be determined whether the observed activation is due to metastasis having occurred, or whether the propensity to

metastatic activation falls short of complete metastasis. Kidney tumors may have arisen as primary tumors in the kidney (Fig. 1e), but be more prone to evolve towards metastatic tumors than their external counterparts (Fig. 5c, d). Systematic phylogenetic/ phylogenomic analysis of numerous tumors upon the same individual should be conducted.

**Tumor transcriptional biomarkers of FP patient outcome**. Finally, to identify putative prognostic biomarkers for patient rehabilitation outcome, tumor transcriptomes from cases with poor outcomes (deceased and euthanized patients, 23 tumor samples) were compared (DESeq2) with those of patients with good outcomes (released, 46 tumor samples). In poor-outcome tumors, 1177 genes were upregulated and 138 genes were downregulated (cutoffs; >±2 log₂ fold change and $P < 0.05$). DE genes were enriched for genes associated with immune and apoptotic functioning, with these genes tending to be downregulated in poor-outcome tumors. To identify strong candidate biomarkers, the top 20 upregulated and downregulated genes were assessed for those showing consistency of expression across both internal and external poor-outcome tumors. This resulted in eleven shortlisted putative biomarkers (Table 1, Fig. 6a). To assess the potential of these shortlisted genes as putative biomarkers, we next examined their expression in an independent study of *C. mydas* FP patients rehabilitated at Sea Turtle Inc, South Padre Island, Texas[55]. Any correlation between patient outcome and expression of the 11 putative biomarkers should represent robust clinically conserved molecular features across different populations of *C. mydas*. Since all animals in the Texas cohort were eventually released, poor-outcome was considered as having over two rounds of surgery, tumor regrowth, and >200 days in rehabilitation, while good outcome was two rounds of surgery or less and <200 days in rehabilitation (Supplementary Data 3). The differential expression of all 11 genes in the Texas cohort recapitulated remarkably well with what was seen in the Florida cohort (Table 1, Fig. 6a), both in terms of directionality (up or downregulation) and magnitude of change (Log₂ fold change). Of these 11 genes, interferon alpha-inducible protein 6 (*Ifi6*) and interferon alpha-inducible protein 27 protein 2B, had by far the highest expression level in good outcome tumors, suggesting they would be readily detectible biomarkers (Fig. 6a and Supplementary Fig. 7a).

## Discussion

Wildlife pathogens have been shown to exacerbate the effects of environmental degradation, habitat loss, and the climate emergency on population levels, potentially leading to local and global extinctions[2,56–59]. As the risk of extinction increases for a given species, the detrimental effects of disease on the population worsen[60]. Anthropogenic activities are stressing habitats[61], and the rapid environmental changes induced by these activities are likely increasing cancer rates in wildlife populations[62]. Human-induced perturbations of inshore marine environments have also been implicated as a co-trigger of the FP tumor epizootic in green sea turtles[10,34,63]. Environmental changes are thought to be key to conferring oncogenicity upon ChHV5-infected turtles, potentially through compromising or modulating the turtles' ability to respond to the viral infection.

We have demonstrated activation of immune-related signaling in FP tumors and shown localized CD3+ T-cell infiltration within new growth tumors. It is interesting that the host still mounts an immune response to ChHV5 within tumors, given the previous links between FP-afflicted turtles and immunosuppression[10,13,63,64]. However, there have been contradictory findings on immunosuppression in FP and non-FP-

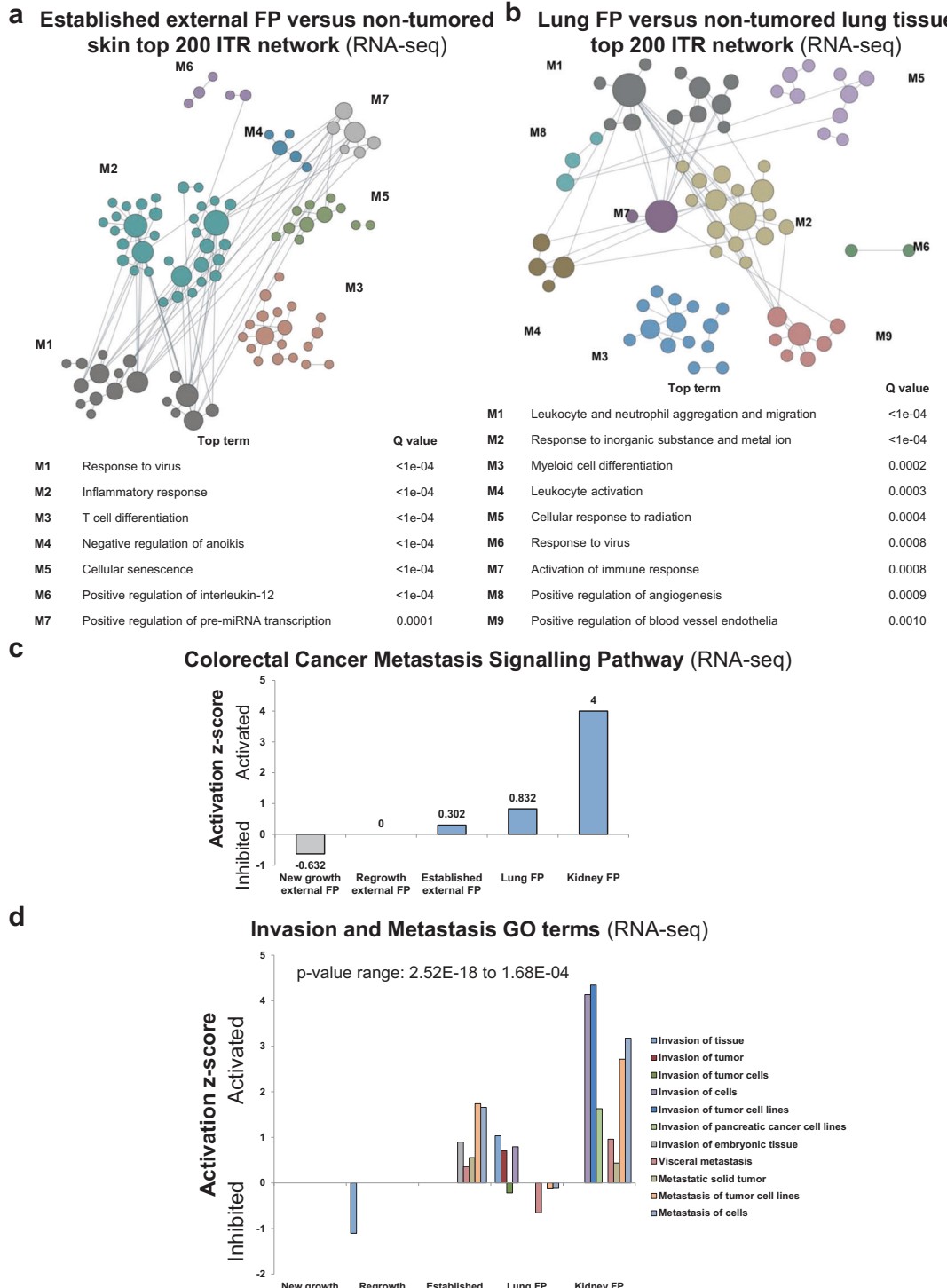

**Fig. 5 Network analysis of inferred transcriptional regulator analysis of fibropapillomatosis tumors.** Network-based functional module discovery of the top 200 ranked ITRs (called by IPA) of **a** established external and **b** lung tumors. **c** Activation/inhibition $z$ scores of the 'Colorectal Cancer Metastasis Signaling' gene ontology (GO) term associated with transcripts differentially expressed in different types of fibropapillomatosis tumors (kidney FP, lung FP, external FP) when compared to their respective non-tumored tissue sources, as detected by IPA. **d** Activation/inhibition $z$ scores of 11 invasion and metastasis-associated GO terms from transcripts differentially expressed in different types of fibropapillomatosis tumors (kidney FP, lung FP, external FP) when compared to their respective non-tumored tissue sources, as detected by IPA.

afflicted individuals, and it is currently not clear whether immunosuppression is a cause or consequence of FP[10,12,13,65,66]. Our transcriptomics and immunohistochemistry show that lymphocytes do infiltrate and mount an immune response within FP tumors, likely against ChHV5, although the immune response appears to be strongest in early-stage new growth tumors. It is important to elucidate why such a response is not sufficient to prevent initial tumorigenesis or prevent continued tumor growth,

**Table 1 Shortlisted putative predictive biomarkers for rehabilitation outcome, as detected by RNA-seq of tumor tissue.**

| Transcript | GFF symbol | GFF description | Florida cohort | | Texas cohort | |
|---|---|---|---|---|---|---|
| | | | Log$_2$ fold change | padj | Log$_2$ fold change | padj |
| Gene18158 | *Pycard* | Apoptosis-associated speck protein containing a CARD | 3.03 | 1.34E−46 | 2.00 | 3.47E−06 |
| Gene17537 | LOC102936464 | Cyclic GMP-AMP synthase | 5.15 | 2.47E−35 | 3.73 | 7.68E−09 |
| Gene15121 | LOC102947706 | Proteasome subunit alpha type-6 | 2.97 | 2.93E−25 | 2.86 | 1.85E−14 |
| Gene11831 | *Ifi6* | Interferon alpha-inducible protein 6 | 4.92 | 1.23E−22 | 5.68 | 1.08E−43 |
| Gene3090 | LOC102943254 | Uncharacterized protein LOC102943254 | 3.051 | 8.86E−21 | 5.53 | 1.65E−26 |
| Gene17471 | LOC102940704 | Interferon alpha-inducible protein 27 protein 2B | 4.65 | 1.21E−20 | 5.22 | 1.01E−20 |
| Gene2107 | LOC102940281 | Uncharacterized protein LOC102940281, partial | 3.86 | 1.00E−19 | 5.59 | 5.25E−31 |
| Gene17278 | *Batf2* | Basic leucine zipper transcriptional factor ATF 2 | 2.94 | 4.81E−19 | 3.75 | 2.11E−21 |
| Gene7946 | LOC102934626 | Interferon alpha-inducible protein 27 protein 2 | 3.08 | 1.07E−18 | 5.00 | 7.84E−23 |
| Gene12380 | *Myh6* | Myosin-6 | 3.75 | 1.74E−18 | 3.86 | 7.28E−07 |
| Gene11311 | LOC102934712 | GTPase IMAP family member 3, partial | 6.33 | 4.81E−18 | 5.82 | 2.16E−12 |

Florida cohort: good outcome is release, whereas poor outcome is died/euthanized in care. Texas cohort (all turtles released): good outcome is two rounds of surgery or less, along with 200 days in rehabilitation.
*GFF* general feature format, *padj*-adjusted *P* value, *CARD* C-terminal caspase-recruitment domain, *ATF* activating transcription factor family, *IMAP* immunity-associated proteins.

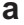

**Putative outcome biomarkers, RNA-seq**
(Good outcome tumors Vs poor outcome tumors)

**Fig. 6 Predictive outcome biomarkers. a** Expression levels of four transcripts differentially expressed between tumors of good outcome and poor-outcome patients, as detected by RNA-seq. Florida cohort: N = 69 samples. Per outcome: released (good outcome) = seven turtles; died/euthanized (poor outcome) = five turtles. Texas cohort: N = 25 samples. Per outcome: prolonged rehabilitation (poor outcome) = two turtles; short rehabilitation (good outcome) = one turtle.

why the response lessens over time, and whether the spontaneous tumor regression that occasionally occurs in non-heavily afflicted FP individuals is a consequence of elevated T-lymphocyte infiltration and reactivation[10,12,25,67].

Immune-related and apoptotic-related genes were downregulated in tumors from poor-outcome patients compared with those of good outcome patients, suggesting that inhibition of these genes may be an important factor in impairing immune and apoptotic anti-cancer responses. Genes, such as *Ifi6* represent potential predictive biomarkers, capable of discriminating between poor and good prognoses in FP-afflicted cases. *Ifi6* is induced by interferon and plays a critical role in the regulation of apoptosis. The biomarkers investigated here likely hold clinical utility for outcome prediction, given that they showed consistency across FP patients originating from different juvenile populations (Texas and Florida), rehabilitated at different facilities, and with RNA extraction and sequencing conducted by independent teams.

The tumor transcriptional profiling reported here found signatures associated with metal, inorganic substances, viruses, and radiation. One of the viral signatures identified was 'Kaposi sarcoma-associated herpesvirus infection'. Kaposi sarcoma is a human cancer which arises in immunocompromised patients, with lesions developing on skin, lymph nodes, or other organs, and is associated with human herpesvirus 8 (HHV8)[3]. Like HHV8, ChHV5 is similarly postulated to drive FP tumor formation in sea turtles also affected by an external environmental or immunomodulatory trigger, a hypothesis which is strengthened by the viral[35] and immune response signaling dynamics revealed by the transcriptomics. Taken together, the transcriptional signatures detected in FP tumors suggest that exposure to UV radiation, metal and/or other inorganic and organic substances may contribute to immunosuppression and subsequent viral-induced oncogenesis.

We show that the host molecular drivers of external and internal FP tumors are largely transcriptionally distinct. Our whole transcriptome profiling indicated that internal tumors are more closely related to the tissue type in which they are found (presumptive tissue of origin), suggesting that they arise from the de novo transformation of these tissues and not as disseminated metastases from external tumors. However, skin cancer-related pathways/GO terms were called for internal tumors from their DE genes (RNA-seq). In addition, there was a trend towards increasing activation of metastasis-associated signaling, particularly for kidney tumors. Mutational analyses comparing multiple tumors within the same individual would elucidate whether certain internal FP tumors are the result of metastatic spread, or whether they merely have tendencies towards the activation of metastasis-associated genes without having arisen from metastatic processes.

No treatments exist for internal tumors, with current practice (including Florida Fish and Wildlife Conservation Commission guidelines) indicating euthanasia for such turtles, regardless of their health status otherwise[68]. Furthermore, surgical excision of external FP tumors often results in high rates of tumor regrowth/recurrence[13,28,36,69]. Therefore, it is imperative that chemotherapeutic approaches are developed to augment surgical removal of external tumors and to provide first-line therapy for applicable internal tumors. Transcriptomic profiling of FP tumors revealed that MAPK, Wnt and TGFβ pathway inhibitors are putative therapies for both external and internal FP, which could be targeted simultaneously to reduce tumor burdens (e.g., systemic targeting via oral dosing). Although kinase inhibitors have been used extensively in human cancers, their application in veterinary medicine is mostly restricted to dogs and cats and only incidentally in other animals[70]. While Wnt signaling is heavily implicated in a wide range of human cancers, no clinically

approved therapeutic yet exists, although clinical trials are ongoing[71,72]. Similarly, TGFβ targeting therapeutics are undergoing human clinical trials[73]. In human head and neck squamous cell carcinoma, infection with human papilloma virus (HPV) increases DNA damage. HPV infection is associated with a loss of TGFβ signaling which increases patient responsiveness to treatment with either radiation or cisplatin[74]. The MAPK/ERK pathway is involved in cell proliferation and survival and is implicated in carcinogenesis[75]. As such, a number of MAPK pathway drugs have been developed, such as MEK and BRAF inhibitors which are most commonly used to treat human melanoma. MAPK/ERK inhibitors have greatly improved the life expectancy of patients with malignant melanoma, but acquired resistance almost inevitably occurs. In addition, longer-term adverse effects of BRAF/MEK inhibitors can occur[76], like skin toxicities that also include secondary malignancies, such as keratoacanthoma and cutaneous squamous cell carcinoma[77]. RAF inhibitors may also cooperate with HPV to promote the initiation of cutaneous tumors[78,79]. Given the complexities of patient responses to MAPK targeting therapeutics, other common pathways between internal and external tumors could be prioritized, or therapeutics selected to separately target external and internal FP tumors.

The key signaling drivers of sea turtle FP revealed here show remarkable similarity to the most prominent pan-cancer drivers of human tumors, as detected by WGS of 2550 tumors from 38 human cancer types[80,81]. This cross-species conserved oncogenic signaling suggests that FP research may help inform human cancer research, but more crucially that sea turtle FP can benefit greatly from advances in therapeutic treatments arising from human oncology.

Our results provide genome-level evidence for the complex relationship between external, new growth, established, post-surgical regrowth, and internal visceral tumors (Table 2). They reveal the host immune responses mounted within FP tumors, and host immunity impairment in tumors from patients with poor outcomes. The application of precision oncology and genomic approaches can assist in determining the molecular events underpinning FP tumor development, and enable the rational design of novel therapeutic interventions (such as pharmacological disruption of MAPK and TGFβ oncogenic signaling) and clinical management strategies. Importantly, the adoption of such approaches can elucidate the specific triggers of FP and the precise mechanisms through which these viral and environmental triggers drive the ongoing FP disease epizootic in sea turtles.

## Methods

### Tissue sampling

*Florida cohort.* Sampling was carried out under permit number MTP-20–236 from the Florida Fish and Wildlife Conservation Commission and with the ethical approval from the University of Florida's Institutional Animal Care and Use Committee (IACUC). All samples were obtained from juvenile green turtles (*C. mydas*), see Supplementary Data 1 for the sex identification of animals whose sex could be determined. External FP tumors were surgically removed and punch biopsies taken of non-tumored areas[28]. New growth tumors were defined as those that arose while the animal was in captivity, were weeks old and small in size (approximately <1 cm max length), while established tumors were present at admission, and large in size (>5 cm max length). Internal tissue samples (tumor and nontumor tissue) were obtained from animals during necropsies conducted immediately after euthanasia. All internal tumors in the study were classified as fibromas. Note that no animal was euthanized for the purposes of this study, but current protocols in Florida for rehabilitating sea turtles determined to harbor internal tumors include euthanasia, as no treatment yet exists for internal tumors, and additional complications arising from surgery and other health concerns sometimes necessitate the humane euthanasia of sea turtles in rehabilitation. Internal tissue samples were treated the same as the external samples. All samples were obtained from juvenile *C. mydas*, as this life stage is the most commonly afflicted by the disease. Sex is not readily determinable in juveniles, but was provided for individuals that were euthanized due to internal tumors or other

**Table 2 Summary of key study findings and their potential future translational applicability.**

| Key points | Future translational applicability |
|---|---|
| Transcriptomics approaches can provide rapid insights into oncogenic drivers of FP. | As for human cancer research, determining the signaling events driving tumor growth is key to the targeted development of improved therapeutic approaches[7,28,33]. |
| The oncological signaling events driving FP tumors show remarkable similarity to the recently revealed human pan-cancer drivers. | FP tumor growth is driven by the same host oncogenic pathways as human cancers. Increased likelihood that therapeutics developed to treat human cancers will be effective in treating FP tumors. |
| Internal FP tumors are molecularly distinct from the more common external tumors. However, there are a small number of conserved potentially therapeutically targetable molecular vulnerabilities in common between internal and external tumors. | Implications for understanding FP pathogenesis and for future treatment strategies of external and internal FP. |
| Turtles attempt to mount an immune response to either the tumor and/or the virus, but in some animals this appears to be insufficient to prevent tumor development and growth. | Foundations for future studies examining the link between these specific immune-related genes and FP biology (e.g., individual susceptibility to FP tumor growth, and likelihood of spontaneous regression or aggressive regrowth). |
| Determined that putative biomarkers (including immune-related genes) are predictive of patient outcomes. | Lead to the development of clinical patient stratification assays (e.g., RT-qPCR- or ELISA-based), to guide rehabilitation decisions. |
| Putative biomarker expression patterns are consistent between distant FP-afflicted populations (Texas and Florida) | Suggests that clinical therapies and biomarkers trialed in one population will be equally effective in other populations. |

complications and in which necropsies were performed, or for individuals that were endoscoped (KARL STORZ, multi-purpose rigid endoscope for small animals) as part of their rehabilitative care (see Supplementary Data 1). Samples were stored until extraction in RNA-later (Qiagen) at −80 °C, according to manufacturer's instructions.

*Texas cohort.* Sampling was carried out under permit number TE181762-4 from the US Fish and Wildlife Service and with the ethical approval of the University of Texas Rio Grande Valley's IACUC. During rehabilitation of stranded sea turtles, external FP tumors were surgically removed using $CO_2$ laser resection. Samples were collected into RNA-later (Qiagen) and stored at −80 °C until processing. The samples reported in the current study correspond to 25 FP tumor samples, from three juvenile green turtles (*C. mydas*) of unknown sex that were sampled as part of an ongoing study of FP in South Texas[55].

**RNA extraction, library preparation, and sequencing from tissue samples**
*Florida cohort.* For RNA-seq samples, total RNA was extracted using either an RNeasy Fibrous Tissue kit (Qiagen, Cat No. 74704) or RNeasy Plus kit (Qiagen, Cat No. 74134) with column-based genomic DNA removal, according to manufacturer's instructions. Ninety RNA samples, comprising 70 FP tumor samples and 20 nontumor samples from 12 juvenile green turtles that stranded in Northern Florida, were used for sequencing. Samples were further categorized by tissue type, as well as growth profile for the external tumors only (see Supplementary Data 1). Sequencing libraries were generated from 500 ng of total RNA using the NEBNext Ultra RNA Library Prep Kit for Illumina (New England Biolabs, Cat No. E7530), including polyA selection, according to manufacturer's protocol. Size and purity of the libraries were analyzed on a Bioanalyzer High Sensitivity DNA chip (Agilent). The RNA samples used for library construction had a RIN value range of 7.2–9.8, with the median RIN value of all samples being 9.1. Libraries were sequenced as paired-end reads with a read length of 100 bp on a HiSeq 3000 (Illumina). ERCC Spike-In Mix (ThermoFisher) was used as an internal control: 2 μL of 1:400 diluted ERCC Spike-In Mix with 500 ng of total RNA input.

*Texas cohort.* Tissue samples were homogenized using a rotor-stator homogenizer and total RNA was extracted using the Qiagen AllPrep DNA/RNA Mini kit, according to the manufacturer's instructions. RNA quality was assessed on a 2200 TapeStation System (Agilent Technologies) with sample RIN values ranging from 6.4 to 9.6 ($\mu = 8.4$). mRNA sequencing libraries were generated from up to 250 ng of total RNA and were prepared using a KAPA RNA HyperPrep kit (Roche Diagnostics) with polyA selection and indexed using a KAPA Dual-Indexed adapter kit (Roche Diagnostics), according to manufacturer's instructions. Prepared libraries were evaluated via the 2200 TapeStation System and were pooled for paired-end 100 bp sequencing on a HiSeq 2500 system (Illumina).

**Quality control and read trimming**
*Florida cohort.* The software FastQC (https://www.bioinformatics.babraham.ac.uk/projects/fastqc/) was used to assess data quality. Reads were then trimmed with trim_galore (The Babraham Institute, version 0.5.0) to remove ends with a Phred quality score less than 30, to remove adaptor sequences, and to remove sequences fewer than 25 bp after trimming. For any samples that contained overrepresented sequences according to FastQC, the trimmomatic tool[82] (version 0.36) was then used to remove these sequences from reads and any sequences less than 25 bp after trimming. The number of raw reads per sample and reads remaining after trimming can be found in Supplementary Data 1.

*Texas cohort.* Sequence data were demultiplexed using bcl2fastq. Raw sequencing reads were processed with Trim Galore (version 0.6.5, https://www.bioinformatics.babraham.ac.uk/projects/trim_galore/), using Cutadapt[83] to remove adapter sequences and indexes from reads and to exclude low-quality sequences using a Phred score of 30 and discarding reads with lengths shorter than 25 bp, unpaired reads were not retained. FastQC (version 0.11.9, (https://www.bioinformatics.babraham.ac.uk/projects/fastqc/)) was used to assess the data quality of the processed reads.

**Read alignment and read counts**
*Florida cohort.* Reads for all RNA-seq samples were then aligned to the draft genome for *C. mydas* [GenBank assembly accession number: GCA_000344595.1][84] using HISAT2[85] (version 2.0.4). The overall alignment rate to the green turtle genome for RNA-seq samples was 82 ± 7% (mean ± SD) (Supplementary Data 1). One sample, an external established growth FP tumor, had an extremely low alignment rate of 26% to the green turtle genome, and was therefore removed from further analysis.

Transcript abundance for *C. mydas* specific transcripts was generated using htseq-count[86] (version 0.6.1p1) with the following parameters: not strand-specific, feature type 'gene', and union mode for *C. mydas* specific transcripts. Count tables for these transcripts were merged for all RNA-seq samples and counts were normalized for gene length and sequencing depth by transcripts per million (Supplementary Data 2).

*Texas cohort.* The CheMyd_1.0 reference assembly for *C. mydas* was obtained from NCBI [GenBank assembly accession number: GCA_000344595.1][84]. Reads were aligned to the reference assembly using HISAT2[85] (version 2.2.0) with an overall alignment per sample ranging from 75.26 to 88.29% ($\mu = 84.00\%$). Transcript abundance was quantified in each sample using htseq-count[86] (version 0.11.2) at the gene level according to the defined genomic features of the CheMyd_1.0 assembly.

**Differential expression analysis**
*Florida cohort.* Prior to differential expression analysis, the raw counts were processed with the RUVseq Bioconductor package[87] (version 0.99.1) using the RUVs method to remove low abundance genes, normalize the RNA-seq data, and remove unwanted variation among replicates. PCA plots (see Fig. 1d) were generated using the PtR script in the Trinity toolkit[88] both before and after RUVseq normalization. The RUVseq-processed matrix was then used to identify DE transcripts using the run_DE_analysis.pl script for the DESeq2 Bioconductor package[89] and available through the Trinity toolkit[88]. The run_DE_analysis.pl script was adjusted to also filter out low abundance genes by removing genes with a mean count ≤10 across all samples prior to differential expression analysis. The resulting lists of DE genes were sorted and filtered to include only those transcripts with an adjusted *P* value of <0.05 and a log₂ fold change of >2 or ≤2. A list of upregulated and downregulated transcripts that overlapped from different sample types was generated and used to create area-proportional Venn diagrams of overlap using BioVenn[90].

Boxplots were generated using BoxPlotR[91]. Volcano plots were generated using Degust: interactive RNA-seq analysis (http://degust.erc.monash.edu/)[92]. For outcome analysis based on *Crabp2* expression (Fig. 2b), the data were not normally distributed (even after log transformation), therefore nonparametric tests (Mann–Whitney U test) were performed ($n = 89$: 20 nontumor, 23 poor-outcome tumor and 46 good outcome tumor samples). Significance was considered to be $p$ values ≤ 0.05.

*Texas cohort*. Differential expression analysis was conducted with DESeq2[89]. Count data generated by htseq-count was imported into a DESeq2DataSet object. Genes with low counts were filtered from the analysis, retaining 17,542 genes with more than five counts in two or more samples. To calculate factors of unwanted variation in this sample, a first differential expression analysis was conducted comparing tumor samples from the two turtles with poor outcomes to the tumor samples from the turtle with good outcomes. There were 4365 genes that were not significantly DE between the two groups with a test statistic $P$ value > 0.5 and these were used to generate a set of empirical control genes for analysis with RUVSeq[87] to estimate two factors of unwanted variation in the data. These factors were then incorporated into the DESeq2 analysis design and an analysis was conducted to identify DE genes with a log$_2$ fold change threshold of 1 and a false discovery rate threshold of 0.05. Previously identified candidate outcome genes (Florida cohort) were then specifically examined in this sample set.

**Pathway analysis and annotation**. Gene lists were analyzed for overrepresented pathways, biological functions, and upstream regulators using IPA (Ingenuity Systems, Qiagen). The $P$ values reported for IPA results were generated by IPA using a right-sided Fisher exact test for over-representation analysis, Benjamini–Hochberg correction for multiple hypothesis testing, and a $z$ score algorithm for upstream analysis; $P$ values < 0.05 were considered significant. For the systems-level analysis, only *C. mydas* DE transcripts that could be annotated to their closest characterized human homolog were included as input.

To better annotate DE transcripts that had turtle-specific gene identifiers [GenBank assembly accession number: GCA_000344595.1], which cannot be used with IPA, the sequence file containing all amino acid sequences for the green turtle genome was re-annotated using PANNZER2 with the –PANZ_FILTER_PERMISSIVE option[93]. When protein and product descriptions for the annotated DE transcripts agreed between PANNZER2 and the original green turtle genome annotation, the PANNZER2 annotation was used if it provided the name of the closest characterized human homolog instead of a turtle-specific identifier. However, it was often the case that the protein and product descriptions for the annotated DE transcripts were not in agreement, so a random subset of the protein sequences of 11 genes was blasted (blastp, https://blast.ncbi.nlm.nih.gov/Blast.cgi?PAGE = Proteins) against the NCBI nonredundant protein database (nr) to determine which annotation method was most accurate. Since 10 out of the 11 protein sequences tested had the original green turtle genome annotation as the top hit, the genome annotation was used for instances in which the two annotations disagreed. If there was no human homolog available in this case, the genome protein description was checked against the STRING database for human homologs[94]. Human annotation was used to enable the most comprehensive systems-level analysis, as human genes have been the most extensively annotated and characterized. Out of all of the unique transcripts identified as DE in all pairwise comparisons, 63% were annotated using the available green turtle genome, 18% were annotated using the PANNZER2 re-annotation of the green turtle genome amino acid sequences, 13% were annotated using the protein description and the STRING database, and 6% of the transcripts either remained unannotated or the annotation was too ambiguous to use in downstream analyses. Interaction networks were generated using STRING[94] (https://string-db.org/), and functional module discovery networks were generated using HumanBase (https://hb.flatironinstitute.org/).

**Histology methodology, embedding, sectioning, and staining**. Turtle tissue samples were surgically removed using a $CO_2$ laser and stored in 4% paraformaldehyde at 4 °C overnight. Samples were washed twice in 1x PBS for 10 min; once in Milli-Q $H_2O$ for 10 min; twice in 50% ethanol for 15 min; twice in 90% ethanol for 15 min; twice in 100% ethanol for 15 min. Samples were stored in 100% ethanol at 4 °C for three nights. Samples were washed one in 100% aniline for 1 h; once in 50:50 aniline:methyl salicylate for 1 h; and twice in 100% methyl salicylate for 1.5 h. Samples were stored in 50:50 methyl salicylate:paraffin at 60 °C overnight. Samples were washed twice in 100% paraffin at 60 °C for 3 h. Samples were stored in 100% paraffin overnight and then embedded in 100% paraffin and stored at 4 °C.

Paraffin blocks were sectioned into 6 μm ribbons of six on charged Fisherbrand Superfrost Plus microscope slides using an AO Spencer "820" microtome and stored at room temperature.

Tissue sections were rehydrated by a series of washes: xylene A for 10 min; xylene B for 5 min; 50:50 xylene:alcohol for 5 min; 100% alcohol A for 5 min; 100% alcohol B for 5 min; 95% alcohol A for 5 min; 95% alcohol B for 5 min; 80% alcohol for 5 min; 70% alcohol for 5 min; 50% alcohol for 5 min; and distilled $H_2O$ for 5 min. Sections were then stored in 1x PBS. Tissue sections were incubated at room temperature for 1.5 h in 200 μl PBS preincubation medium (1% normal goat serum + 0.1% albumin solution from bovine serum + 0.1% Tritonx100 + 0.02% sodium azide + PBS). Tissue sections were incubated at room temperature overnight in primary antibody medium or control medium (1:100 primary anti β-catenin

antibody from rabbit Sigma C2206 + 1:100 primary anti β-actin antibody (cytoskeleton marker) from mouse Sigma A5441 + PBS preincubation medium, or PBS preincubation medium only, respectively). Tissue sections were washed twice in 1x PBS for 20 min. Tissue sections were incubated at 37 °C for 2 h in 1:250 FITC GAR (goat anti-rabbit) + TRITC GAM (goat anti-mouse) + PBS preincubation medium. Secondary antibodies were from Jackson ImmunoResearch Labaoratires Inc. (West Grove, PA) and were affinity purified and selected for very low cross-reactivity with other animal sources of Ig. Tissue sections were washed with 300 ml 1x PBS and 2 μl Hoechst 33342, trihydrochloride, trihydrate (Life Technologies Corp., Eugene OR) for 10 min. Tissue sections were washed twice in 1x PBS for 10 min. Three drops of 60% glycerol in PBS containing PPD (p-phenylenediamine, 0.3 mg/ml) as a fluorescence quench inhibitor were applied to the sections and a cover slip then added to each slide (6 tissue sections per slide). A Leica SP5 confocal microscope was used to visualize and capture images of the fluorescent staining in each tissue section.

For CD3 staining, tissue sections were sent to the University of Florida Veterinary Diagnostic Laboratories core facility, and were stained with rabbit anti-human CD3 ε chain antibody clone LN10 (RM-9107-S1 Thermosfisher, Labvision) and an alkaline-phosphatase based red chromogen detection kit and co-stained with hematoxylin. This CD3 antibody has previously been validated as also specifically recognizing green sea turtle CD3[50].

**Retinoic acid therapeutic methodology**. Photos with a scale bar were taken of patients undergoing ectopic RA treatment using an Olympus Tough TG-5, bi-weekly, for the duration of their treatment. This allowed the surface area of each tumor to be analyzed using ImageJ. Direct measurements were also taken bi-weekly using iGaging digital calipers to record the length and width of each tumor. A topical RA therapeutic (Spear Tretinoin Cream 0.1%) was applied for a 6–8-week course depending on the veterinary determination of patient status. Each treated tumor was coupled with a control tumor in the same anatomical location on the opposite side of the body. Tumor length, width, and surface area were analyzed to determine the overall effectiveness of topical RA treatments for inhibiting FP tumor growth.

**Statistics and reproducibility**. Details regarding, sample size, statistical tests employed, and replicates are presented in each relevant section above.

**Reporting summary**. Further information on research design is available in the Nature Research Reporting Summary linked to this article.

## Data availability

The RNA-Seq data including raw reads are deposited in NCBI (https://www.ncbi.nlm.nih.gov/) under BioProject ID: PRJNA449022 (https://www.ncbi.nlm.nih.gov/bioproject/PRJNA449022).

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

## Acknowledgements

Funding was generously provided by The Sea Turtle Conservancy, Florida Sea Turtle Grants Program under project number 17-033R, the Save Our Seas Foundation under project number SOSF 356, the National Save The Sea Turtle Foundation, Inc. under project name Fibropapillomatosis Training and Research Initiative, and a Welsh Government Sêr Cymru II and the European Union's Horizon 2020 research and innovation program under the Marie Skłodowska-Curie grant agreement no. 663830-BU115. This research was also supported by Gumbo Limbo Nature Center, Inc d/b/a Friends of Gumbo Limbo (a 501c3 non-profit organization) through a generous donation through their Graduate Research Grant program and by an Irish Research Council Government of Ireland Postgraduate Scholarship, under project number GOIPG/2020/1056. Maximilian Stammnitz is funded by a Gates Cambridge PhD scholarship. The Texas study was funded by the University of Texas Rio Grande Valley Transforming Our World Strategic Plan. Warmest thanks to Mark Q. Martindale, Nancy Condron and the veterinary and rehabilitation staff and volunteers of the Sea Turtle Hospital at Whitney Laboratories. The Texas study would particularly like to acknowledge the involvement of the following project contributors: Marcelo Leandro, Ignacio Martinez Escobedo, Juan M. Peralta, Jeff George, Dr. Brian Stacey, Dr. John Blangero, Dr. Megan Keniry, and Dr. Joanne E. Curran. Thanks also are due to David Moraga, Yanping Zhang, and Mei Zhang of UF's Interdisciplinary Center for Biotechnology Research Core Facilities and Nicole Stacy and the UF College of Veterinary Medicine Diagnostic Laboratories and Elizabeth Ryan for informative discussions, and Florida Fish and Wildlife Conservation Commission's Meghan Koperski for valuable assistance with permitting.

## Author contributions

D.J.D. designed and supervised the project. D.J.D, J.A.F., K.Y., P.L., N.B.B. and A.C.L. generated the data. D.J.D., K.Y., J.A.F., N.B.B., L.W., M.R.S., A.K., R.T., A.C.L., E.P.M. and C.S. performed data and bioinformatics analysis. B.B., D.R.R, C.B.E., R.T., M.A.D., N.N. and T.W.M. provided veterinary care, including tumor removal surgeries and diagnostics, and led and conducted the RA treatments. D.J.D., K.Y. and J.A.F. wrote the manuscript; L.W., N.B.B., C.S., M.R.S., A.K., E.P.M., S.C., G.C., A.K. and J.W. performed critical reading of the manuscript. All authors read and approved the final manuscript.

## Competing interests

The authors declare no competing interests.

## Additional information

[1]The Whitney Laboratory for Marine Bioscience and Sea Turtle Hospital, University of Florida, St. Augustine, FL 32080, USA. [2]Department of Biology, University of Florida, Gainesville, FL 32611, USA. [3]Department of Human Genetics, School of Medicine, University of Texas Rio Grande Valley, Brownsville, TX, USA. [4]South Texas Diabetes and Obesity Institute, School of Medicine, University of Texas Rio Grande Valley, Brownsville, TX, USA. [5]Department of Biological Sciences, School of Natural Sciences, Faculty of Science and Engineering, University of Limerick, Limerick, Ireland. [6]Transmissible Cancer Group, Department of Veterinary Medicine, University of Cambridge, Cambridge CB3 0ES, UK. [7]Systems Biology Ireland & Precision Oncology Ireland, School of Medicine, University College Dublin, Belfield, Dublin 4, Ireland. [8]Molecular Ecology and Fisheries Genetics Laboratory, School of Biological Sciences, Bangor University, BangorGwynedd, LL57 2UW, UK. [9]Sea Turtle Inc., South Padre Island, TX, USA. [10]Gladys Porter Zoo, Brownsville, TX, USA. [11]Present address: Menzies Institute for Medical Research, University of Tasmania, Hobart, Tasmania, Australia. [12]These authors contributed equally: Kelsey Yetsko, Jessica A. Farrell. ✉email: duffy@whitney.ufl.edu

