## [Transparent Peer Review File · Communications Biology]

Reviewers' comments:

Reviewer #1 (Remarks to the Author):

This study combines transcriptomic and immunohistochemical profiling of various types of tumors associated with fibropapillomatosis in green sea turtles. Transcriptomics results deliver putative biomarkers for case prognosis, putative molecular signaling pathways to be targeted with cancer therapeutics, and putative drivers/triggers (endogenous and exogenous) of tumor development, regrowth, and potential for metastasis. Additionally, the paper provides novel, pertinent information about T-cell and innate immune responses to FP tumor development, which can be useful for understanding why certain turtles with ChHV5 infection either do not grow tumors or are able to spontaneously regress mild FP disease. This paper is chock-full of highly valuable data about FP disease, which can and hopefully will lead to multiple spinoff studies in multiple fields of research, to follow up on these preliminary analyses. The manuscript is well-written, although some work can be done to integrate the many different parts of the paper, and to filter the results into a more easily comprehensible central message. The addition of bullet points detailing possible future research, and describing the real-world applicability that can be mounted based on these data would help the reader to fully appreciate the breadth and scope of this study for beginning to understand some of the key 'unknowns' about FP. There is a lot here and it would be useful to further distill some of the key points for the reader. Minor edits and comments are provided in the attached marked-up draft. The researchers appear more than capable of conducting this work and providing skilled interpretations of the results.

-- Annie Page-Karjian, DVM, PhD

Reviewer #2 (Remarks to the Author):

This paper reports a study that has the potential to make significant progress in our understanding of the fibropapilloma tumours in green turtles.

The text contains a number of errors that could have been detected by careful editing and cross-referencing by any one of the authors.

Copied in attachment by Editor

Comments on the paper:

Molecular characterization of a marine turtle tumor epizootic, profiling external, internal and post-surgical regrowth tumors

This paper reports a study that has the potential to make significant progress in our understanding of the fibropapilloma tumours in green turtles.

The text contains a number of errors that could have been detected by careful editing and cross-referencing by any one of the authors.

There are approximately 20 authors on this paper. Prior to publication all of these authors will have signed a declaration that they are satisfied with the text in its present form. It is abundantly clear that the rigour applied to the editing by the authors left something to be desired. I was supplied with a document that was incomplete. It did not have all of the tables. Some of the abbreviations have been duplicated. One of the references was duplicated. If the authors had used a program such as EndNote this could have been avoided. Some of the text quoted tables or figures that were incorrect. It is disappointing to see that authors are happy to be on the paper without putting the work in that is required to ensure that the publication is ready to go when it is submitted.

Specific comments:

Page 3

This is very likely to be disease prevalence not disease incidence.

This page was an introduction which is a critical literature review. It cites two publications, numbers 23 and 27. These two publications are based on a study that used single sets of primers and it would appear that none of the lines that appeared in the gels were sequenced to confirm that viral sequence had been amplified.

I suggest that there are three alternatives. One if you are totally satisfied that the authors have been able to prove beyond doubt that their conclusions are correct then they should be included without modification. The second alternative is to quote these papers with some reservations. The third alternative is to avoid using these papers as there are many other studies that would be quite suitable to support the statements being made.

You will note that the conclusions in these two papers are at odds with your own results (Yetsko et al., 2020).

Page 4

The tumours clearly did not show minimal overlap. The word "show" is an active term suggesting that the tumours participated in demonstrating overlap. This sentence should be revised to indicate that observations were made and I assume what is being said is that they were frequently tumours in both lung and kidney or that the transcripts in both of these tumours were similar.

Page 9

interferon alpha-inducible protein 6 (Ifi6) is defined on this page and then again on page 10.

Is the figure being referred to 6A or 6?

Page 10

Why 6A there is no 6B

Supplemental figure 7A was not labelled as such.

Page 13

Fibropapillomatosis (FP) was defined on page 2

Surely the word required is euthanasia.

Page 15

This is likely to be figure 2A not 1A

Page 17

I am not too sure what bovine albumin serum is. Surely this is just bovine albumin or bovine serum albumin.

Page 20 and 21

References 6 and 9 are duplicates.

Reference number 18 is a relatively obscure report. All of the references that are quoted should be such that the majority of readers can readily obtain access. I doubt that your readers would be able to readily access this particular publication.

Page 24

Reference number 69

This is the publication of a consortium (Consortium, 2020)

Pages 29 and 30

Revise the numbering of the supplementary figures noted as six.

The final supplemental figure Interferon alpha-inducible protein 27 protein 2B does not have a label.

CONSORTIUM, I. T. P.-C. A. O. W. G. 2020. Pan-cancer analysis of whole genomes. *Nature*, 578, 82-93.

YETSKO, K., FARRELL, J., STAMMNITZ, M. R., WHITMORE, L., WHILDE, J., EASTMAN, C. B., RAMIA, D. R., THOMAS, R., KRSTIC, A., LINSER, P., CREER, S., CARVALHO, G., BURKHALTER, B., MURCHISON, E. P., SCHNITZLER, C. & DUFFY, D. J. 2020. Mutational, transcriptional and viral shedding dynamics of the marine turtle fibropapillomatosis tumor epizootic. *bioRxiv*,

2020.02.04.932632.

Reviewer #3 (Remarks to the Author):

Molecular characterization of a marine turtle tumor epizootic, profiling external, internal and post-surgical regrowth tumors Kelsey Yetsko^{1^}, Jessica Farrell^{1,2^}, Nicholas B. Blackburn^{3,4}, Liam Whitmore^{1,5}, Maximilian R. Stammnitz⁶, Jenny Whilde¹, Catherine B. Eastman¹, Devon Rollinson Ramia¹, Rachel Thomas¹, Aleksandar Krstic⁷, Paul Linser¹, Simon Creer⁸, Gary Carvalho⁸, Mariana Devlin⁹, Nina Nahvi⁹, Ana C. Leandro^{3,4}, Thomas W. deMaar¹⁰, Brooke Burkhalter¹, Elizabeth P. Murchison⁶, Christine Schnitzler^{1,2} and David J. Duffy^{1,2,5,7,8*}. Communications Biology COMMSBIO-20-1547

Overview: Authors did transcriptomics and IHC of new, established, regrowth, lung and kidney tumors from FP affected turtles in Florida and Texas. This paper reads like someone threw a bunch of data out there hoping something would stick. It is so poorly thought out, poorly presented, and disorganized that one is hard pressed to see what, if anything, is of use to increase our understanding of FP. Authors would do well to really think through what they are trying to do and come up with a more coherent message grounded in reality and what we know about FP. As written, this MS sheds no light on FP in sea turtles. Some specifics follow.

Page 2, first paragraph of intro: Please provide a definition for "precision oncology". Do you mean that RNASeq is somehow more precise than other approaches and if so how? One might argue that gross exams can be pretty precise as can microscopic pathology or IHC. Simply listing genes expressed and then speculating on how they influence disease without additional confirmatory experiments seems just as imprecise or precise as other approaches no? For instance, how exactly is understanding transcriptomics of tumors enhancing our understanding of why FP is increasing in some areas and not others?

Page 3, ph 1: Might want to add here that FP has been declining in Hawaii (1)

P3, ph 2: "Even so, ChHV5 infection alone is not sufficient to induce FP tumor growth." Should add this citation as well (2)

P3, ph 2, last sentence. The paper cited as 30 is the same one I'm currently reading right now.

P 3: "Determining the contribution of each facet of this multifactorial disease will be key to combatting this anthropogenic-implicated disease epizootic both at the level of individual clinical treatment and for population level management/mitigation strategies." What exactly do you mean here? What facets? Do you mean understanding the different molecular signals in different tumor types? I can understand how this could inform treatment, but how is that useful for management/mitigation? And how does that address the virus?

Page 4, ph1: That FP is anthropogenically mediated is overstated. FP is most often documented where people look (3), so there will by this very fact be an association with human. Obviously, this is very difficult to disentangle, but this needs to be acknowledged. There is no question that, like any other animal diseases which are caused by a confluence of agent, host, and environment, FP has an environmental component, but how much of that is due to humans is open to question.

P4, ph2: There is no Table S1 presented in the review materials. You cite Figure 1B for transcript

overlap, but Fig. 1B is gross photos of turtles. Also, your tumors need clearer definitions. One assumes that new growth tumors were tumors that arose while animal was in captivity while established tumors were those that were present at admission? In the upper right panel of Figure 1, you say this is a regrowth tumor, but I do not see a tumor at all. It is not evident at all from CAT scan that this is a tumor. Much more convincing would be a gross photograph. Moreover, the lungs in this cat scan are clear. Where are the lung tumors? Finally, internal tumors are typically characterized further histologically as fibromas, myxofibromas, or fibrosarcomas of low grade malignancy (4-5). This would seem important from gene expression standpoint. How were internal tumors you analyzed characterized?

P4, Ph 3: Again, no concordance between figure cited in text and narrative. Figure 1C shows cat scan of perfectly clear lung (no tumors). There is no Table S2.

P5, ph 1: What do you define as a positive outcome? Survival to release?

P 5, ph 2: We know that tumors are cancerous from gross and microscopic pathology. It would stand to reason, therefore, that cancer gene ontology terms would be prominent no? This whole exercise seems a bit tautologic. There is also a lot of non-sensical narrative. "The remaining two non-cancer GO terms related to muscle and were inhibited, likely indicating an absence of muscle in FP tumors compared with non-tumored skin controls." It is well known from histology that FP tumors are mainly connective tissue and fibroblasts (4-6), so I'm not sure how transcriptomics telling me there is no muscle there is informative.

P5, Ph 3: Again, it is well known that external tumors have prominent lymphoid infiltrates (4, 6), so no surprise that one would see gene expression for lymphoid processes. How is that informative?

P 6, ph 1: Given that internal tumors are morphologically distinct from external (4,5) would you not expect gene expression to differ? Why are your results important and how do they shed light on disease pathogenesis?

P6, Ph 2: Actually, looking at Figure 3C, all the CD3 staining is occurring in the stratum basale. This is not the layer where inclusions occur (stratum spinosum) (6-7), thus arguing against response to ChHV5. Indeed, I see no inclusions nor ballooning degeneration in any of the micrographs. Moreover, it is difficult for reader to judge whether CD3 staining was any more prominent in new growth tumors absent data for other tumor types.

P7, ph 1: This makes no sense. You start arguing that retinoic acid would be promising avenue of treatment, then go on to say it would not be presenting a bunch of graphs and hand waving. Which is it? Useful or not? Seems you thought it might be useful because you treated with RA in methods.

P 7, ph 2: In figure 4 D I see no evidence of staining for B Catenin. And in any case, how is that useful or relevant to our understanding of FP pathogenesis?

P8, ph2 : Here you are flooding the reader with a bunch of data with absolutely no relationship to reality of FP as we know it. There is no evidence of HHV8, a human herpesvirus, in turtle FP. This is nonsense.

P8, ph3, P9, ph 1: This makes no sense. Metastasis is a process whereby tumors cells seed distally and establish new tumors. Here, you say that external tumors have repressed metastatic pathways but kidneys are elevated. So presumably you are implying that kidney tumors metastasize to skin? Yet no one has ever documented internal tumors without external tumors. Moreover, you go on later to

say that transcriptomic signal of kidney tumors is closer to kidney than to other tumors and same for lung (close to lung) which actually makes more sense. This would completely torpedo the argument that internal tumors are metastatic.

Discussion

Paragraph 1; Lot of empty verbiage, no surprise with CD3 (see above), it's pretty clear that immunosuppression in turtles is a sequela to FP and not a prerequisite (8-10), and that immune response to the virus varies with location (11). "Immune-related and apoptotic-related genes were downregulated in tumors from poor outcome patients compared with those of good outcome patients, suggesting that inhibition of these genes may be an important factor in impairing immune and apoptotic anti-cancer responses." Does not tell me much. Is it the downregulation of those genes that is promoting tumors or does presence of tumors downregulate genes? Only way to really answer that is to measure gene expression over time. Certainly, what you present here has no predictive value, at least within constraints of experimental design.

Paragraph 2: This is all wild speculation. See above regarding HHV8. No evidence that metals have anything to do with tumors. If UV is an issue, why is FP declining in Hawaii (UV if anything there is likely getting worse).?

Paragraph 3: So which is it? Are internal tumors metastatic or not? Seems that with all these graphs and figures and charts, we are no closer to understanding pathogenesis of FP.

Paragraph 4: Hard to see how any of these therapies can be realistically implemented in turtles. How useful is any of this?

Paragraph 5: You have not made a case that any of these data are useful to inform FP pathogenesis, treatment, or epidemiology.

Paragraph 6: Delete-empty verbiage.

P13 Ph 1: I was unable to find a table S1.

P15, ph 1: Fig 1a shows turtle tumors, not PCA plots.

P16-17: What was the rationale for staining for B catenin or B actin? After all, the latter is a common house keeping gene present in most cells. How is staining for these informing our understanding of FP?

P 17: Why were tumors treated with retinoic acid? How is that relevant?

Seems like the texas tumor sample processing is not all that different than that for Florida samples. Suggest merge and condense.

References

- 1) Chaloupka, M., G. H. Balazs, and T. M. Work. Rise and fall over 26 years of a marine epizootic in Hawaiian green sea turtles. *J Wildl Dis* 45:1138-1142. 2009.
- 2) Herbst, L., and P. Klein. Green turtle fibropapillomatosis: challenges to assessing the role of environmental cofactors. *Environmental Health Perspectives* 103:27-30. 1995.
- 3) Chaloupka, M., T. M. Work, G. H. Balazs, S. K. K. Murakawa, and R. M. Morris. Cause-specific temporal and spatial trends in green sea turtle strandings in the Hawaiian Archipelago (1982-2003).

Mar Biol 154:887-898. 2008.

- 4) Work, T. M., G. H. Balazs, R. A. Rameyer, and R. Morris. Retrospective pathology survey of green turtles (*Chelonia mydas*) with fibropapillomatosis from the Hawaiian Islands, 1993-2003. *Dis Aquat Org* 62:163-176. 2004.
- 5) Norton, T. M., E. R. Jacobson, and J. P. Sundberg. Cutaneous fibropapillomas and renal myxofibroma in a green turtle, *Chelonia mydas*. *J Wildl Dis* 26:265-270. 1990.
- 6) Herbst, L. H., E. R. Jacobson, P. A. Klein, G. H. Balazs, R. Moretti, T. Brown, and J. P. Sundberg. Comparative pathology and pathogenesis of spontaneous and experimentally induced fibropapillomas of green turtles (*Chelonia mydas*). *Vet Pathol* 36:551-564. 1999.
- 7) Work, T. M., J. Dagenais, G. H. Balazs, N. Schettle, and M. Ackermann. Dynamics of virus shedding and in-situ confirmation of chelonid herpesvirus 5 in Hawaiian green turtles with fibropapillomatosis. *Vet Pathol* 52:1195-1220. 2014.
- 8) Cray, C., R. Varella, G. D. Bossart, and P. Lutz. Altered in vitro immune responses in green turtles (*Chelonia mydas*) with fibropapillomatosis. *J. Zoo Wildl Med* 32:436-440. 2001.
- 9) Work, T. M., R. A. Rameyer, G. H. Balazs, C. Cray, and S. P. Chang. Immune status of free-ranging green turtles with fibropapillomatosis from Hawaii. *J Wildl Dis* 37:574-581. 2001.
- 10) Work, T. M., and G. H. Balazs. Relating tumor score to hematology in green turtles with fibropapillomatosis in Hawaii. *J Wildl Dis* 35:804-807. 1999.
- 11) Work, T. M., J. Dagenais, A. Willimann, G. Balazs, K. Mansfield, and M. Ackermann. Differences in antibody responses against Chelonid alphaherpesvirus 5 suggest differences in virus biology between green turtles from Hawaii and Florida. *J Virol*:JVI.01658-01619. 2020.

Response to Reviewers

We would like to thank the reviewers for their careful consideration of the manuscript and for their thoughtful and constructive comments. We have provided detailed rationale for the few suggestions we disagreed with and have implemented all the other requested changes, which we believe have helped to further improve the manuscript.

Author responses in **red text**.

Reviewers' comments:

Reviewer #1 (Remarks to the Author):

This study combines transcriptomic and immunohistochemical profiling of various types of tumors associated with fibropapillomatosis in green sea turtles. Transcriptomics results deliver putative biomarkers for case prognosis, putative molecular signaling pathways to be targeted with cancer therapeutics, and putative drivers/triggers (endogenous and exogenous) of tumor development, regrowth, and potential for metastasis. Additionally, the paper provides novel, pertinent information about T-cell and innate immune responses to FP tumor development, which can be useful for understanding why certain turtles with ChHV5 infection either do not grow tumors or are able to spontaneously regress mild FP disease. This paper is chock-full of highly valuable data about FP disease, which can and hopefully will lead to multiple spinoff studies in multiple fields of research, to follow up on these preliminary analyses. The manuscript is well-written, although some work can be done to integrate the many different parts of the paper, and to filter the results into a more easily comprehensible central message. The addition of bullet points detailing possible future research, and describing the real-world applicability that can be mounted based on these data would help the reader to fully appreciate the breadth and scope of this study for beginning to understand some of the key 'unknowns' about FP. There is a lot here and it would be useful to further distill some of the key points for the reader. Minor edits and comments are provided in the attached marked-up draft. The researchers appear more than capable of conducting this work and providing skilled interpretations of the results.

-- Annie Page-Karjian, DVM, PhD

Thank you for your detailed comments. All suggested changes in the reviewer's pdf have now been implemented in the revised manuscript.

Regarding the TPM/normalized read count (Fig. 6): the Florida values were calculated and reported in TPM, while the Texas values were calculated and reported in normalized read counts. While these provide comparable relative measurements, the precise bioinformatic process to generate them vary slightly (a factor of the independent analysis of the Florida and Texas datasets).

Thank you for the suggestion; as requested, we have now included a new table (Table 2) summarising the key findings of the manuscript and their relevance for future clinical translation. We believe this helps greatly in contextualising the findings.

Reviewer #2 (Remarks to the Author):

This paper reports a study that has the potential to make significant progress in our understanding of the fibropapilloma tumours in green turtles.

The text contains a number of errors that could have been detected by careful editing and cross-referencing by any one of the authors.

Copied in attachment by Editor

Comments on the paper:

Molecular characterization of a marine turtle tumor epizootic, profiling external, internal and post-surgical regrowth tumors

This paper reports a study that has the potential to make significant progress in our understanding of the fibropapilloma tumours in green turtles.

The text contains a number of errors that could have been detected by careful editing and cross-referencing by any one of the authors.

There are approximately 20 authors on this paper. Prior to publication all of these authors will have signed a declaration that they are satisfied with the text in its present form. It is abundantly clear that the rigour applied to the editing by the authors left something to be desired. I was supplied with a document that was incomplete. It did not have all of the tables. Some of the abbreviations have been duplicated. One of the references was duplicated. If the authors had used a program such as EndNote this could have been avoided. Some of the text quoted tables or figures that were incorrect. It is disappointing to see that authors are happy to be on the paper without putting the work in that is required to ensure that the publication is ready to go when it is submitted.

Thank you for your detailed comments, which we have now implemented. Apologies for any errors in the original submission; these stemmed from the splitting of an earlier version of the manuscript where the host and viral FP results were presented within a single manuscript.

Specific comments:

Page 3

This is very likely to be disease prevalence not disease incidence. Changed to prevalence as suggested.

This page was an introduction which is a critical literature review. It cites two publications, numbers 23 and 27. These two publications are based on a study that used single sets of primers and it would appear that none of the lines that appeared in the gels were sequenced to confirm that viral sequence had been amplified.

I suggest that there are three alternatives. One if you are totally satisfied that the authors have been able to prove beyond doubt that their conclusions are correct then they should be included without modification. The second alternative is to quote these papers with some reservations. The third alternative is to avoid using these papers as there are many other studies that would be quite suitable to support the statements being made.

You will note that the conclusions in these two papers are at odds with your own results (Yetsko et al., 2020).

We have removed references to these papers as requested. They were not crucial to the point we were making, as in all instances these references were cited in conjunction with other related papers. Dryad deposited sequence data does seem to be available for these publications, but a comprehensive review of the merits of these papers is beyond the scope of the current research manuscript, not least as we primarily focus on host rather than viral events.

Page 4

The tumours clearly did not show minimal overlap. The word “show” is an active term suggesting that the tumours participated in demonstrating overlap. This sentence should be revised to indicate that observations were made and I assume what is being said is that they were frequently tumours in both lung and kidney or that the transcripts in both of these tumours were similar. Revised to ‘...tumor transcriptomes had...’.

Page 9

interferon alpha-inducible protein 6 (Ifi6) is defined on this page and then again on page 10. Apologies for the duplicated definition. This has been corrected so that Ifi6 is now only defined upon the first instance of its use in the text.

Is the figure being referred to 6A or 6? Labelling single figure panels (with A, B etc) is the style of some journals. We assumed that was the case for Comms Bio also. If not, we are happy to remove the A.

Page 10

Why 6A there is no 6B. Same as above: Labelling single figure panels (with A, B, etc.) is the

style of some journals. We assumed that was the case for Comms Bio also. If not we are happy to remove the A.

Supplemental figure 7A was not labelled as such. Apologies for this inconsistency. We can either include the A in Fig. 6, or remove it from Supplemental Fig 7, as directed by the editor.

Page 13

Fibropapillomatosis (FP) was defined on page 2 Apologies for the duplication. This has now been corrected.

Surely the word required is euthanasia. Amended.

Page 15

This is likely to be figure 2A not 1A. Thank you, it has now been corrected.

Page 17

I am not too sure what bovine albumin serum is. Surely this is just bovine albumin or bovine serum albumin. To improve clarity, we have altered the wording to 'albumin solution from bovine serum'.

Page 20 and 21

References 6 and 9 are duplicates. Apologies, reference 9 has now been deleted (error occurred due to a duplicated EndNote entry).

Reference number 18 is a relatively obscure report. All of the references that are quoted should be such that the majority of readers can readily obtain access. I doubt that your readers would be able to readily access this particular publication. While we agree that the reference is to a local report, we included it as it is widely available online. For example, the full pdf of this report is easily obtained from publicly available tools such as Google Scholar (see below). Additionally, we have previously cited this report in our other published Comms Bio papers. That being said, we are happy to remove the citation here, if you would prefer.

Page 24

Reference number 69

This is the publication of a consortium (Consortium, 2020) *This format was from the direct import of the citation (to EndNote) from the paper's page on the journal's website (<https://www.nature.com/articles/s41586-020-1969-6#citeas>). But if the reviewer and editor would prefer us to alter the reference to appear as Consortium, then we are more than happy to make this change.*

Pages 29 and 30

Revise the numbering of the supplementary figures noted as six. *Revised.*

The final supplemental figure Interferon alpha-inducible protein 27 protein 2B does not have a label. *Apologies, this has now been corrected.*

CONSORTIUM, I. T. P.-C. A. O. W. G. 2020. Pan-cancer analysis of whole genomes. *Nature*, 578, 82-

93.

YETSKO, K., FARRELL, J., STAMMNITZ, M. R., WHITMORE, L., WHILDE, J., EASTMAN, C. B., RAMIA, D. R., THOMAS, R., KRSTIC, A., LINSER, P., CREER, S., CARVALHO, G., BURKHALTER, B.,

MURCHISON, E. P., SCHNITZLER, C. & DUFFY, D. J. 2020. Mutational, transcriptional and viral shedding dynamics of the marine turtle fibropapillomatosis tumor epizootic.

bioRxiv,2020.02.04.932632.

Reviewer #3 (Remarks to the Author):

Molecular characterization of a marine turtle tumor epizootic, profiling external, internal and post-surgical regrowth tumors Kelsey Yetsko^{1^}, Jessica Farrell^{1,2^}, Nicholas B. Blackburn^{3,4}, Liam Whitmore^{1,5}, Maximilian R. Stammnitz⁶, Jenny Whilde¹, Catherine B. Eastman¹, Devon Rollinson Ramia¹, Rachel Thomas¹, Aleksandar Krstic⁷, Paul Linser¹, Simon Creer⁸, Gary Carvalho⁸, Mariana Devlin⁹, Nina Nahvi⁹, Ana C. Leandro^{3,4}, Thomas W. deMaar¹⁰, Brooke Burkhalter¹, Elizabeth P. Murchison⁶, Christine Schnitzler^{1,2} and David J. Duffy^{1,2,5,7,8*}.
Communications Biology COMMSBIO-20-1547

Overview: Authors did transcriptomics and IHC of new, established, regrowth, lung and kidney tumors from FP affected turtles in Florida and Texas. This paper reads like someone threw a bunch of data out there hoping something would stick. It is so poorly thought out, poorly presented, and disorganized that one is hard pressed to see what, if anything, is of use to increase our understanding of FP. Authors would do well to really think through what they are trying to do and come up with a more coherent message grounded in reality and what we know about FP. As written, this MS sheds no light on FP in sea turtles. Some specifics follow.

We are sorry that the reviewer does not see the value of our work. These comments are in direct contrast to the other two reviewers, e.g. Reviewer 1, 'This paper is chock-full of highly valuable data about FP disease, which can and hopefully will lead to multiple spinoff studies in multiple fields of research' and Reviewer 2, 'This paper reports a study that has the potential to make

significant progress in our understanding of the fibropapilloma tumors in green turtles'. We have made changes to the manuscript to help highlight and contextualise the numerous novel findings described in the manuscript, such as the inclusion of Table 2 as recommended by Reviewer 1.

We and other research groups are building the foundation of FP genomic-era data. As for human cancers, while the initial research is by necessity more explorative, it lays the foundation for future targeted studies to further enhance our understanding of the drivers of FP tumors and ultimately novel therapeutic and management strategies. Already results from this manuscript have identified putative clinical biomarkers of rehabilitation outcome, and identified the underlying oncogenic signalling pathways driving FP tumor growth, for new growth, established, regrowth and internal tumors. FP tumor transcriptomes have never been assessed on this scale before, and no previous -omic data exists for internal, regrowth or new growth tumors. Please see the new Table 2 for the future translation potential for these findings. We also strongly disagree with the assertion that such approaches do not shed light on the drivers of disease, or lead to tangible progress. The power of such approaches is attested to by the near-ubiquitous adoption of -omic technologies for human disease research, diagnosis, patient stratification and therapeutic target identification, by academics, clinical practitioners and industrial researchers. Indeed, our first FP transcriptomics paper¹, published in *Communications Biology*, already demonstrated the clinical translation of transcriptomics to FP treatment, having identified a human anti-cancer drug capable of greatly reducing the incidence of regrowth of surgically removed eye FP tumors.

Page 2, first paragraph of intro: Please provide a definition for "precision oncology". Do you mean that RNASeq is somehow more precise than other approaches and if so how? One might argue that gross exams can be pretty precise as can microscopic pathology or IHC. Simply listing genes expressed and then speculating on how they influence disease without additional confirmatory experiments seems just as imprecise or precise as other approaches no? For instance, how exactly is understanding transcriptomics of tumors enhancing our understanding of why FP is increasing in some areas and not others?

The following definition of precision oncology has now been added: 'Precision oncology incorporates recent advances in -omic technologies (genomics, transcriptomics, proteomics, metabolomics, high-throughput histology/imaging etc.) and computational advancements and applies them to the molecular profiling of tumors to provide mechanistic clarity, and to identify targetable alterations and predictive biomarkers²⁻⁴. Precision oncology is rapidly developing and has entered the mainstream of human clinical practice²⁻⁴.'

We agree that precision medicine is not a panacea (for details please see my review of both the promise and pitfalls of precision medicine³). However, the advances wrought in practically every field of biology by the advent of the genomics-era cannot be refuted. The first studies (such as this one) to apply -omic approaches to FP will not immediately answer all of the unresolved questions about FP disease. However, they do provide new tools and baseline information that over the coming years will enable the field to progress beyond the limits of more traditional approaches, and resolve these questions.

Precision medicine is not set against microscopic pathology. Rather, it encompasses it, and genomics and microscopy are complementary (as demonstrated in this manuscript, e.g.

histological confirmation of CD3 and β -catenin staining in FP tumors). Genomics primarily delivers data averaged across the entire sample, whereas histology can provide valuable spatial information. Pathology is embracing technological advancements wrought by the computational improvements underpinning precision medicine. Precision medicine includes the move towards high-throughput histology/pathology being embraced by human hospitals, including massively-parallel marker staining within individual tissue sections, and the incorporation of machine learning and AI tools applied to histology sections to assist with disease diagnosis.

Transcriptomics enables a measure of the relative contribution of host and viral genes in driving FP tumor growth to be determined. Similarly, it enables the specific host signalling pathways perturbed by environmental exposures to be identified (as outlined in the manuscript). Both of these abilities help to inform future targeted studies into viral genes specifically expressed in FP and transcriptomically shortlisted environmental exposures (based on the tell-tale transcriptional changes within FP tumors). However, as discussed in the manuscript the combination of field experiments coupled to mutational signature analysis (whole genome sequencing of FP and patient-matched non-tumor tissue) is a next logical step for deciphering the specific environmental factors which have induced changes in FP genomes (i.e. tell-tale mutational patterns linked to specific exposures/carcinogens).

Page 3, ph 1: Might want to add here that FP has been declining in Hawaii (1)

We have now added the Hawaii information to the paragraph.

P3, ph 2: "Even so, ChHV5 infection alone is not sufficient to induce FP tumor growth." Should add this citation as well (2)

Thank you, we have now included the suggested reference.

P3, ph 2, last sentence. The paper cited as 30 is the same one I'm currently reading right now.

The reviewer is correct, reference 30 is the pre-print associated with this manuscript. Parts of the manuscript appear in the pre-print, although not the entire manuscript. For example, the putative outcome biomarker data and the Texas cohort data are not in the pre-print. Primarily the main difference is that this Comms Bio manuscript deals with the host-related aspects of the preprint, the reference to the pre-print is a place holder and will be replaced with a reference to the sister viral paper (associated with the pre-print) once that paper has been published.

P 3: "Determining the contribution of each facet of this multifactorial disease will be key to combatting this anthropogenic-implicated disease epizootic both at the level of individual clinical treatment and for population level management/mitigation strategies." What exactly do you mean here? What facets? Do you mean understanding the different molecular signals in different tumor types? I can understand how this could inform treatment, but how is that useful for management/mitigation? And how does that address the virus?

The facets referred to are the host, viral and environmental drivers of the disease, including their interplay at the molecular level. Knowledge of all 3 facets is required to inform management strategies (from quarantine during rehab, to potential population level mitigation strategies). For the viral-related findings from our studies and more in-depth discussion on those please see the preprint (reference 30 of the manuscript).

Page 4, ph1: That FP is anthropogenically mediated is overstated. FP is most often documented where people look (3), so there will by this very fact be an association with human. Obviously, this is very difficult to disentangle, but this needs to be acknowledged. There is no question that, like any other animal diseases which are caused by a confluence of agent, host, and environment, FP has an environmental component, but how much of that is due to humans is open to question.

To better reflect the current uncertainty we have removed the term 'anthropogenic-implicated'. We agree that more research is needed regarding the specific environmental co-factor(s) of FP, and only once it/they have been causally confirmed can the true extent of human involvement be determined. However, there is correlative evidence of FP being more prevalent in areas with higher human activity than areas with lower human activity, for example please see Van Houtan et al. 2010⁵.

P4, ph2: There is no Table S1 presented in the review materials. You cite Figure 1B for transcript overlap, but Fig. 1B is gross photos of turtles. Also, your tumors need clearer definitions. One assumes that new growth tumors were tumors that arose while animal was in captivity while established tumors were those that were present at admission? In the upper right panel of Figure 1, you say this is a regrowth tumor, but I do not see a tumor at all. It is not evident at all from CAT scan that this is a tumor. Much more convincing would be a gross photograph. Moreover, the lungs in this cat scan are clear. Where are the lung tumors? Finally, internal tumors are typically characterized further histologically as fibromas, myxofibromas, or fibrosarcomas of low grade malignancy (4-5). This would seem important from gene expression standpoint. How were internal tumors you analyzed characterized?

Apologies, Table S1 was originally absent from the single pdf submitted, it was subsequently provided to the journal.

Figure 1B is correctly cited, it shows transcript overlap. Figure 1A is the gross photos of turtles. If the reviewer is suggesting that they would like to see each individual image making up panel 1A having its own label we are happy to consider that.

Yes, the reviewer is correct, new growth tumors arose while the animal was in captivity, are weeks old and are small in size (~<1cm max length), while established tumors were present at admission, and large in size (>5cm max length). This information has now been added to the methods.

Figure 1 regrowth tumors were confirmed as such by a commercial veterinary histopathology diagnostic service. Additionally, the new growth tumor shown in Fig.1 had very high ChHV5 loads (DNA load assessed by qPCR), whereas the matching patient skin samples did not. Please see pre-print for viral load information (reference 30 of the manuscript).

The CAT scan tumor shown was identified by our veterinarian, and subsequently confirmed by necropsy and histopathology. The individual shown did not have lung tumors, but had a large kidney tumor, as indicated in the figure legend.

All internal tumors in the study were fibromas, as independently determined by a commercial veterinary histopathology laboratory service. To make this information more accessible it has now been added to the methods section.

P4, Ph 3: Again, no concordance between figure cited in text and narrative. Figure 1C shows cat scan of perfectly clear lung (no tumors). There is no Table S2.

Figure 1C shows the overlap of external and internal differentially expressed transcripts. The CAT scan is part of Fig.1A, i.e. all of the gross turtle images. If the panel labelling is not on the pdf received by the reviewer please let us know and we will amend this. The labels are clearly present on the version of the pdf we download (during validation) from the journal's submission site.

Alternatively, if the reviewer is asking for each individual image to have its own panel label, then that is something we are more than willing to discuss, although it would also have implications for all other figures (i.e. CD3 and β -catenin staining panels).

P5, ph 1: What do you define as a positive outcome? Survival to release?

Yes, as described in the manuscript text (P5, ph 1) and in the figure legend, positive outcome was defined as release. ‘...*Crabp2* expression across our turtle patient cohort. Patients with positive outcome (release) tended to have tumors with lower *Crabp2* expression when compared with patients that died in care or were euthanized due to advanced disease (Fig. 2B),...’

P 5, ph 2: We know that tumors are cancerous from gross and microscopic pathology. It would stand to reason, therefore, that cancer gene ontology terms would be prominent no? This whole exercise seems a bit tautologic. There is also a lot of non-sensical narrative. "The remaining two non-cancer GO terms related to muscle and were inhibited, likely indicating an absence of muscle in FP tumors compared with non-tumored skin controls." It is well known from histology that FP tumors are mainly connective tissue and fibroblasts (4-6), so I'm not sure how transcriptomics telling me there is no muscle there is informative.

We have consistently been requested by a number of reviewers of previous papers/grants not to refer to FP as cancerous. Surprisingly, it would appear that FP being cancerous is still a debated concept in the FP field. Therefore, these gene ontology findings help to cement the view that FP tumors are indeed cancerous.

Regarding the absence of muscle, we agree that this is a well-established fact, and we do not suggest that this is a novel discovery. Rather, it highlights that the transcriptomics reflects the underlying biology of FP, thus validating the approach. As the bioinformatic approaches correctly

call the absence of muscle in FP tumors (with no *a priori* knowledge/input), this supports the validity of the novel findings (oncogenic pathway/GO terms etc) found in this approach.

This was contextualised by the next sentence ‘The skin controls are sampled by punch-biopsies, so they also contain subcutaneous tissue.’ However, to make this point more clearly we have amended the section to now read as follows: "The remaining two non-cancer GO terms related to muscle and were inhibited. This finding helps to validate the analysis pipeline (GO term results were called without the input of any *a priori* FP knowledge) as FP tumors are known to contain less muscle tissue than non-tumored skin punch biopsies which also contain subcutaneous tissue." The citations to reviewer’s references 4-6 have been added.

P5, Ph 3: Again, it is well known that external tumors have prominent lymphoid infiltrates (4, 6), so no surprise that one would see gene expression for lymphoid processes. How is that informative?

It is informative as we not only generated information to say that there is lymphoid infiltration, but the transcriptomics reveal exactly which genes are involved (i.e. not all infiltration and immune responses are identical). **Crucially, specific immune-related genes identified by the transcriptomics were even able to putatively predict patient outcome.**

Furthermore, the transcriptomics is fully quantitative, not having potential discrepancies between individuals scoring slides (e.g. see agreement in transcriptome results between the independent Florida and Texas cohorts). Genome-wide transcriptomic data can help identify the precise immune mechanisms involved and inform future potential therapeutic strategies relating to immune checkpoint targeting therapies (currently being assessed as anti-cancer therapeutics for human tumors). Additionally, this quantitative data is the first to reveal the precise differences between the different tumor types studied. Similarly, the CD3 data reveals changes between the tumor types, and the precise spatial location of CD3 within the tumors.

P 6, ph 1: Given that internal tumors are morphologically distinct from external (4,5) would you not expect gene expression to differ? Why are your results important and how do they shed light on disease pathogenesis?

Expectation is not the same as scientific validation. Without empirical evidence (like that presented in the manuscript), there is no way of testing our expectations.

As outlined in the manuscript, another widely held assumption in the FP field is that internal tumors are all primary, and not the result of metastatic spread. Despite its prevalence, this assumption has not been tested and no molecular information previously existed about the relationship of internal versus external FP. This is a major unresolved question regarding FP pathogenesis. Again, our data not only confirm that they are transcriptionally distinct, but reveal the specific genes and pathways that are divergent, which is important to understand the origin and progression of these tumors. Our data are the first ever evidence that internal tumors have a propensity towards the activation of metastatic signalling pathways.

It is also important in relation to the future development of internal tumor-specific biomarkers (e.g. ELISA-based assessments), which would be a useful rehabilitation tool as many

rehabilitation facilities (especially those outside developed countries) do not have routine access to CAT scans (current internal tumor diagnosis approach).

Additionally, when the host drivers of internal FP are combined with our ChHV5 transcriptomic data the importance (or otherwise) of ChHV5 in the establishment of internal tumors can start to be resolved.

P6, Ph 2: Actually, looking at Figure 3C, all the CD3 staining is occurring in the stratum basale. This is not the layer where inclusions occur (stratum spinosum) (6-7), thus arguing against response to ChHV5. Indeed, I see no inclusions nor ballooning degeneration in any of the micrographs. Moreover, it is difficult for reader to judge whether CD3 staining was any more prominent in new growth tumors absent data for other tumor types.

Indeed, this is why in the manuscript we state that the immune response could be in relation to viral infection, but equally may be occurring in response to the tumor cells themselves. Immune evasion is one of the key hallmarks of cancer^{6,7}. We have now removed the following qualification: *'the latter being more likely given the localization pattern'*. Given our ChHV5 transcriptomic findings, the perceived prominence of ChHV5 expression in driving new growth, established, regrowth external FP and internal tumors is called into question.

Regarding data from other tumor types, these are present in Supplemental Fig 3A.

P7, ph 1: This makes no sense. You start arguing that retinoic acid would be promising avenue of treatment, then go on to say it would not be presenting a bunch of graphs and hand waving. Which is it? Useful or not? Seems you thought it might be useful because you treated with RA in methods.

We have amended that section of text in the manuscript slightly to assist with conveying the intended message. Also, please find a more in-depth description below.

Retinoic acid response genes are activated in 3 types of FP tumors (Fig. 4C). As such, retinoid inhibition would be a potential treatment strategy for FP, but retinoid treatment (activation) would not. In human cancer, retinoid therapy (activation) is extremely effective against some cancer types, while the same therapy is ineffective or increases tumor growth rates in other cancer types. Therefore, knowing the status of retinoic acid signalling in a tumor is key to understanding whether this cost-effective therapy is likely to be beneficial. The transcriptomics in this paper suggest retinoid therapy will not be effective against FP, i.e. retinoid treatment would further activate the RA-associated signalling already occurring in FP.

Prior to having the RNA-seq results we followed the response of FP tumors to retinoid therapy (Supplemental Fig. 3B), and retinoid therapy was not effective. In the absence of the transcriptomics we had evaluated RA, because if it was effective it would have provided an accessible and inexpensive treatment option.

These retinoid therapy results were included in this manuscript (Supplemental Fig. 3B) as they confirm the transcriptomic findings. Similar to the downregulation of muscle-associated GO terms (see above), the consistency between the transcriptomics and the tumor treatment results

helps to validate the transcriptomic dataset, suggesting that the anti-cancer therapeutics identified by the transcriptomics (e.g. MAPK inhibitors etc.) would be effective in treating FP. As shown in Fig. 4C, the MAPK signalling pathway is activated, and the response genes to MAPK inhibitors are suppressed in FP. Therefore, treatment with MAPK inhibitors would likely block this MAPK pathway activation, inhibiting oncogenic proliferation.

Unfortunately, the top candidate drugs from this analysis are designer therapies which are cost-prohibitive in a rehabilitation hospital setting. As these drugs come off-patent their use as anti-FP treatments should be evaluated.

P 7, ph 2: In figure 4 D I see no evidence of staining for BCatenin. And in any case, how is that useful or relevant to our understanding of FP pathogenesis?

All green staining (cell membrane and nuclear) in Fig. 4D and Supplemental Fig. 4A, B is t3-catenin protein.

Cancer is a cellular disease driven by aberrant cell signalling^{6,7}. Understanding the signalling events driving tumor pathogenesis has been crucial to our ability to understand and treat human and animal cancers. Understanding of the underlying cellular signalling enables the identification of outcome biomarkers and targeted prevention and treatment strategies. Ignoring the host signalling events in FP is akin to intentionally keeping the field in the pre-molecular era of cancer research. Wnt/t3-catenin signalling is one of the top oncogenic pathways in human and animal cancers, and it is commonly used to segregate patients for treatment regimens and to accurately predict patient outcomes. Upregulation of t3-catenin signalling has been well documented in human cancers as it leads to cell proliferation and predisposes cells to tumorigenesis. Due to their prominence as oncogenic drivers, intense efforts have been made to develop anti-Wnt/t3-catenin therapies, including a number of compounds currently undergoing clinical trial. By revealing the oncogenic drivers of FP we can determine which are shared with human tumors. Therefore, FP research and treatment can immediately benefit from the wealth of knowledge and anti-cancer therapies that exists regarding human cancer, for those conserved oncogenic drivers.

P8, ph2 : Here you are flooding the reader with a bunch of data with absolutely no relationship to reality of FP as we know it. There is no evidence of HHV8, a human herpesvirus, in turtle FP. This is nonsense.

We never claimed that HHV8 or any other human virus is present in FP.

What the FP transcriptomics shows is that genes known to be involved in host response to viral infection are differentially expressed in FP. This does not mean that human viruses are in FP tumors, nor did we ever make such claims. It shows that host viral response genes known to be involved with other viral infections are differentially expressed in FP, not what specific viruses are present. To identify what viruses are present from the transcriptomics one must align the reads against viral genomes (see pre-print⁸, or Duffy et al. 2018¹), or conduct metagenomics analysis.

The most advanced analysis knowledge bases (e.g. IPA, String etc.) are primarily populated with data from humans (and rodents). This is why host viral response gene pathways identified from these databases tend to include the names of human viruses. But these host genes are shared by other vertebrates (including sea turtles), so what is being revealed by the analysis is that host genes with conserved involvement in mediating host responses to viral infection are differentially expressed (again not what species-specific virus is driving these changes).

P8, ph3, P9, ph 1: This makes no sense. Metastasis is a process whereby tumors cells seed distally and establish new tumors. Here, you say that external tumors have repressed metastatic pathways but kidneys are elevated. So presumably you are implying that kidney tumors metastasize to skin? Yet no one has ever documented internal tumors without external tumors. Moreover, you go on later to say that transcriptomic signal of kidney tumors is closer to kidney than to other tumors and same for lung (close to lung) which actually makes more sense. This would completely torpedo the argument that internal tumors are metastatic.

The reviewer is misinterpreting the data. These data in no way suggest that external tumors are seeded from internal ones (see clustering of skin and external tumor types in Fig. 1E). Were it the case that external tumors originated from the metastatic process, they would have a higher activation score of metastatic signalling pathways.

We have included the following sentence in the manuscript to make it even clearer that we never suggest that the external tumors are a result of metastatic spread from internal tumors:

‘Kidney tumors may have arisen as primary tumors in the kidney (Fig. 1E), but may be more prone to evolve towards metastatic tumors than their external counterparts.’

Metastasis is an open question, one that requires detailed lineage tracing of multiple tumors in the same individual, which is achievable by analyzing the mutational spectrum of multiple tumors. The transcriptomics results here show that there is a possibility that internal tumors have arisen metastatically, or are capable of seeding other tumors, but equally they may only have a tendency towards the activation of metastatic signalling pathways, without ever crossing the metastatic threshold. For instance, kidney tumors may have arisen as primary tumors in the kidney (see Fig. 1E), but be more prone to evolve towards metastatic tumors than their external counterparts. The transcriptomics does not resolve whether metastasis has occurred, and further investigation at the DNA level is required, especially the comparison of multiple kidney tumors within the same individual.

In any event, having metastatic pathways activated could be a sign that they originate from metastatic spread, as much as one that they are able to seed other tumors. Additionally, if they could seed tumors it is highly possible that they might seed other internal tumors but not external tumors. It is well documented that different cancer types have a propensity to only seed metastatic tumors in specific tissue types (due to tissue micro-environment factors). For most cancer types, metastasis cannot spread tumors equally to every tissue type.

Regarding the reviewer’s comment: Yet no one has ever documented internal tumors without external tumors. Just as a point of interest, this is extremely rare, but has been observed, though

has not been published (personal communication Brian A. Stacy, University of Florida and NOAA). Regardless, as explained above we are not suggesting that external tumors are seeded from internal tumors.

Discussion

Paragraph 1; Lot of empty verbiage, no surprise with CD3 (see above), it's pretty clear that immunosuppression in turtles is a sequela to FP and not a prerequisite (8-10), and that immune response to the virus varies with location (11). "Immune-related and apoptotic-related genes were downregulated in tumors from poor outcome patients compared with those of good outcome patients, suggesting that inhibition of these genes may be an important factor in impairing immune and apoptotic anti-cancer responses." Does not tell me much. Is it the downregulation of those genes that is promoting tumors or does presence of tumors downregulate genes? Only way to really answer that is to measure gene expression over time. Certainly, what you present here has no predictive value, at least within constraints of experimental design.

Re 'it's pretty clear that immunosuppression in turtles is a sequela to FP and not a prerequisite' This is still a somewhat debated topic in the FP field (e.g. reviewed in Jones et al. 2016⁹) and environmentally-induced immunosuppression is still a leading theory as to why some turtles (in the same geographic location) are susceptible to FP and others are not. Additional research (including detailed time course analysis) will help to resolve this question.

Re '...measure gene expression over time'. We agree with the reviewer, future detailed time-course analysis would be most informative, especially with regards to causation. The comparison of early and later stage FP tumors in this manuscript is a first step towards disentangling the molecular progress of the disease.

The results present here have predictive value for patient outcome, not for whether the altered gene expression is a cause or consequence of FP. Outcome prediction is an important issue in FP rehabilitation, as the rehabilitation process is cost- and effort-intensive. Any tools to help predict patient outcome in advance can feed into clinical decision making and assist with decisions regarding the distribution of limited resources to the cases where they can achieve the most good. The putative biomarkers revealed by this study represent the first step towards clinically applicable biomarker development. Already the selected markers showed utility in the independent Texas dataset.

Paragraph 2: This is all wild speculation. See above regarding HHV8. No evidence that metals have anything to do with tumors. If UV is an issue, why is FP declining in Hawaii (UV if anything there is likely getting worse).?

Again, we do not say HHV8 is present, but that host genes known to be involved in driving the response to herpesvirus infection (e.g. HHV8) are differentially expressed in FP. The same is true re. UV and metals: the transcriptomics say that host genes known to alter their expression to these exposures are differentially expressed in FP. That these genes are altered is hard evidence, not

speculation. However, it is not proof that these exposures are the causative factor of FP. To ascertain that, in-depth investigation is required. What the transcriptomics do is show which exposures there is evidence for at the transcriptional level (i.e. limiting the number of potential exposures that are likely to play a role in FP tumorigenesis), but additional experimentation is required to investigate each exposure.

Regarding metals and FP, without getting into the relative merits of the study, metals have been postulated to have a role in FP by some researchers, e.g. da Silva et al. 2016¹⁰.

Regarding Hawaii, to scientifically evaluate the case for Hawaii it would require a detailed study to answer that question including examining any confounding factors (depth of water, differences in viral strains¹¹, culling strategy in Hawaii etc.). Furthermore, while declining in 2014 (the most recent date for published data to be available) 44% of stranded turtles in Hawaii (not too degraded to be assessed) had FP¹². That is not a negligible rate.

Like any research, this manuscript's evidence-based assessment of FP generates a wealth of results worthy of further in-depth study.

Paragraph 3: So which is it? Are internal tumors metastatic or not? Seems that with all these graphs and figures and charts, we are no closer to understanding pathogenesis of FP.

Here we have revealed a previously unknown propensity towards the activation of metastatic signalling pathways in kidney FP tumors. This is a novel result, which argues in favor of future in-depth assessment of whether some FP tumors arise due to metastatic spread. The knowledge that this is a possibility (i.e. changes in metastatic-associated gene expression) is important and helps to highlight this question as a future research priority.

Paragraph 4: Hard to see how any of these therapies can be realistically implemented in turtles. How useful is any of this?

The same could be said of FP viral-related work, with decades of research producing no therapies, while, in fact, we have already shown that FP host transcriptomics results can be translated into novel therapies and clinical success¹. The similarity of FP oncogenic signalling and that of humans (this manuscript) and the effectiveness of human anti-cancer drugs in preventing FP tumor regrowth¹, combine to show that as a field, FP genomic-medicine research has only just scratched the surface of therapy advancement.

Please see the new Table 2, for the potential clinical relevance of the manuscript's main findings.

Paragraph 5: You have not made a case that any of these data are useful to inform FP pathogenesis, treatment, or epidermiology.

We have previously shown how such -omic approaches can identify drugs to improve clinical treatment of FP eye tumors, preventing post-surgical regrowth¹. In the current manuscript we reveal the molecular drivers of internal, new growth and regrowth FP, as well as a greatly

expanded cohort of external tumors. This wealth of data will be utilised in further longer-term studies to similarly identify candidate chemotherapeutic treatments for FP (see the new Table 2).

Similarly, our identification of outcome biomarkers lays the groundwork for future studies beyond the clinic and may help identify why some individuals in free-ranging populations experience high tumor burdens, while other turtles in the same locations are relatively unaffected (a long standing and as yet unresolved epidemiological question).

Historically, the role of host oncogenic signalling in maintaining and driving FP tumor growth has been understudied, with decades of research on the ChHV5 virus leaving us no closer to new non-surgical treatment options. Our data confirm that the role of host genes should not be ignored in FP, especially as FP tumors are driven by the very same oncogenic signalling pathways that drive human and other animal tumors. The conserved oncogenic signalling of FP is underscored by the similarities in FP transcriptomes from geographically distant populations (Florida and Texas). Our data show that altered host gene expression is key to FP pathogenesis and requires equal consideration to viral and environmental factors associated with the disease.

Paragraph 6: Delete-empty verbiage.

Deleted.

P13 Ph 1: I was unable to find a table S1.

Apologies, due to its size Table S1 was not included as part of the original single pdf submission, but was subsequently provided to the journal.

P15, ph 1: Fig 1a shows turtle tumors, not PCA plots.

Apologies, amended to Fig. 1D.

P16-17: What was the rationale for staining for B catenin or B actin? After all, the latter is a common house keeping gene present in most cells. How is staining for these informing our understanding of FP?

t3-actin was used precisely because it is a common housekeeping gene, i.e. a ubiquitous marker of the cytoskeleton. As such, t3-actin has long been used as a marker of cellular actin frameworks. Just as DAPI or Hoechst is used to visualise nuclei, t3-actin enables the visualisation of cell cytoplasm and boundaries (via their actin framework). The combination of nuclear and nonnuclear cell staining enables the cellular localisation of t3-catenin (our oncogene of interest) to be confirmed. To improve clarity, we have added the term 'Cytoskeleton' to the t3-actin staining in Fig 4D.

Wnt/t3-catenin is one of the a main oncogenic signaling pathways¹³⁻¹⁶. t3-catenin can be activated in tumors through a variety of processes (including overexpression of t3-catenin or Wnt ligands, activating t3-catenin mutation, or inactivating mutations of t3-catenin repressors such as APC). In addition to quantitative readouts of t3-catenin mRNA, proteins and target gene transcriptional signatures, staining to reveal t3-catenin's spatial distribution and sub-cellular localization is an extensively utilized means of identifying cell populations (in development or cancer) with active

nuclear β -catenin. Such cells are generally highly proliferative and can drive rapid tumor growth. Understanding β -catenin activity in FP (both at the transcriptional and cellular level) can reveal the underlying oncogenic signaling drivers of tumor growth and can identify future candidate treatments (targeted therapeutics). Irrespective of the environmental or viral triggers of FP, it is the resulting expression changes in host genes that drive tumor growth. Therefore, near-term improvements to rehabilitation success (including prevention of post-surgical tumor regrowth) are most readily achieved by identifying the specific host oncogenic drivers (such as Wnt/ β -catenin signaling) and improving our treatment repertoire.

P 17: Why were tumors treated with retinoic acid? How is that relevant?

Please see above in the response to Reviewer 3 Result section comments.

Retinoic acid (RA) is a commonly utilized, cost-effective anti-cancer treatment. Prior to having the transcriptomic results RA was assessed for its effectiveness in treating FP tumors. The relevance, and the reason it was included in this manuscript, is that subsequently the RNA-seq revealed that RA response genes are activated in FP, therefore further activating these genes (by application of RA) was unlikely to reverse the tumor phenotype. Therefore, the treatment results (no major reduction of tumor growth) help to validate that the transcriptomics accurately reflect the underlying signalling events driving FP.

The role of retinoic acid signalling in cancer is complex. In some cancer types it helps drive tumor growth, whereas in others it is an effective treatment inducing cancer cell differentiation and apoptosis. Whether RA is an effective treatment or a promotor of tumor growth depends on the underlying RA signalling pathway dynamics of the tumor type. For FP these underlying signalling dynamics have been revealed by the transcriptomics, and the treatment results are in agreement. This is informative as it not only helps to reveal the oncogenic signalling pathways driving FP tumor growth, and helps to validate the transcriptomic data, but also suggests that treatments inhibiting retinoic acid signalling might be effective therapeutics for FP tumors.

Seems like the Texas tumor sample processing is not all that different than that for Florida samples. Suggest merge and condense.

The Texas and Florida methods information have now been re-arranged for easier comparison.

References

- 1) Chaloupka, M., G. H. Balazs, and T. M. Work. Rise and fall over 26 years of a marine epizootic in Hawaiian green sea turtles. *J Wildl Dis* 45:1138-1142. 2009.
- 2) Herbst, L., and P. Klein. Green turtle fibropapillomatosis: challenges to assessing the role of environmental cofactors. *Environmental Health Perspectives* 103:27-30. 1995.
- 3) Chaloupka, M., T. M. Work, G. H. Balazs, S. K. K. Murakawa, and R. M. Morris. Cause-specific temporal and spatial trends in green sea turtle strandings in the Hawaiian Archipelago (1982-2003). *Mar Biol* 154:887-898. 2008.
- 4) Work, T. M., G. H. Balazs, R. A. Rameyer, and R. Morris. Retrospective pathology survey of green turtles (*Chelonia mydas*) with fibropapillomatosis from the Hawaiian Islands, 1993-2003.

Dis Aquat Org 62:163-176. 2004.

- 5) Norton, T. M., E. R. Jacobson, and J. P. Sundberg. Cutaneous fibropapillomas and renal myxofibroma in a green turtle, *Chelonia mydas*. *J Wildl Dis* 26:265-270. 1990.
- 6) Herbst, L. H., E. R. Jacobson, P. A. Klein, G. H. Balazs, R. Moretti, T. Brown, and J. P. Sundberg. Comparative pathology and pathogenesis of spontaneous and experimentally induced fibropapillomas of green turtles (*Chelonia mydas*). *Vet Pathol* 36:551-564. 1999.
- 7) Work, T. M., J. Dagenais, G. H. Balazs, N. Schettle, and M. Ackermann. Dynamics of virus shedding and in-situ confirmation of chelonid herpesvirus 5 in Hawaiian green turtles with fibropapillomatosis. *Vet Pathol* 52:1195-1220. 2014.
- 8) Cray, C., R. Varella, G. D. Bossart, and P. Lutz. Altered in vitro immune responses in green turtles (*Chelonia mydas*) with fibropapillomatosis. *J. Zoo Wildl Med* 32:436-440. 2001.
- 9) Work, T. M., R. A. Rameyer, G. H. Balazs, C. Cray, and S. P. Chang. Immune status of free-ranging green turtles with fibropapillomatosis from Hawaii. *J Wildl Dis* 37:574-581. 2001.
- 0) Work, T. M., and G. H. Balazs. Relating tumor score to hematology in green turtles with fibropapillomatosis in Hawaii. *J Wildl Dis* 35:804-807. 1999.
- 1) Work, T. M., J. Dagenais, A. Willimann, G. Balazs, K. Mansfield, and M. Ackermann. Differences in antibody responses against Chelonid alphaherpesvirus 5 suggest differences in virus biology between green turtles from Hawaii and Florida. *J Virol*:JVI.01658-01619. 2020.

References as part of author responses:

- 1 Duffy, D. J. *et al.* Sea turtle fibropapilloma tumors share genomic drivers and therapeutic vulnerabilities with human cancers. *Communications Biology* **1**, 63, doi:10.1038/s42003-018-0059-x (2018).
- 2 Garraway, L. A., Verweij, J. & Ballman, K. V. Precision Oncology: An Overview. *Journal of Clinical Oncology* **31**, 1803-1805, doi:10.1200/jco.2013.49.4799 (2013).
- 3 Duffy, D. J. Problems, challenges and promises: perspectives on precision medicine. *Briefings in Bioinformatics* **17**, 494-504, doi:10.1093/bib/bbv060 (2016).
- 4 Schwartzberg, L., Kim, E. S., Liu, D. & Schrag, D. Precision Oncology: Who, How, What, When, and When Not? *American Society of Clinical Oncology Educational Book*, 160-169, doi:10.1200/edbk_174176 (2017).
- 5 Van Houtan, K. S., Hargrove, S. K. & Balazs, G. H. Land Use, Macroalgae, and a Tumor-Forming Disease in Marine Turtles. *PLoS ONE* **5**, e12900, doi:10.1371/journal.pone.0012900 (2010).
- 6 Hanahan, D. & Weinberg, R. A. The Hallmarks of Cancer. *Cell* **100**, 57-70 (2000).
- 7 Hanahan, D. & Weinberg, R. A. Hallmarks of cancer: the next generation. *Cell* **144**, doi:10.1016/j.cell.2011.02.013 (2011).
- 8 Yetsko, K. *et al.* Mutational, transcriptional and viral shedding dynamics of the marine turtle fibropapillomatosis tumor epizootic. *bioRxiv*, 2020.2002.2004.932632, doi:10.1101/2020.02.04.932632 (2020).
- 9 Jones, K., Ariel, E., Burgess, G. & Read, M. A review of fibropapillomatosis in Green turtles (*Chelonia mydas*). *The Veterinary Journal* **212**, 48-57, doi:<http://dx.doi.org/10.1016/j.tvjl.2015.10.041> (2016).
- 10 da Silva, C. C., Klein, R. D., Barcarolli, I. F. & Bianchini, A. Metal contamination as a possible etiology of fibropapillomatosis in juvenile female green sea turtles *Chelonia mydas* from the southern Atlantic Ocean. *Aquatic Toxicology* **170**, 42-51, doi:<http://dx.doi.org/10.1016/j.aquatox.2015.11.007> (2016).

- 11 Work, T. M. *et al.* Differences in Antibody Responses against Chelonid Alphaherpesvirus 5 (ChHV5) Suggest Differences in Virus Biology in ChHV5-Seropositive Green Turtles from Hawaii and ChHV5-Seropositive Green Turtles from Florida. *Journal of Virology* **94**, e0165801619, doi:10.1128/jvi.01658-19 (2020).
- 12 Hargrove, S., Work, T., Brunson, S., Foley, A. M. & Balazs, G. Proceedings of the 2015 International Summit on Fibropapillomatosis: Global Status, Trends, and Population Impacts. 85 (2016).
- 13 Anastas, J. N. & Moon, R. T. WNT signalling pathways as therapeutic targets in cancer. *Nat Rev Cancer* **13**, 11-26, doi:http://www.nature.com/nrc/journal/v13/n1/supinfo/nrc3419_S1.html (2013).
- 14 Reya, T. & Clevers, H. Wnt signalling in stem cells and cancer. *Nature* **434**, 843-850 (2005).
- 15 Polakis, P. Wnt signaling and cancer. *Genes & Development* **14**, 1837-1851, doi:10.1101/gad.14.15.1837 (2000).
- 16 Duffy, D. J. *et al.* Wnt signalling is a bi-directional vulnerability of cancer cells. *Oncotarget* **7**, 60310-60331 (2016).

REVIEWERS' COMMENTS:

Reviewer #1 (Remarks to the Author):

The manuscript is much improved by the revisions. A few more suggested revisions are below:

Abstract

L42: change to "immunohistochemical"

L53: define 'poor outcomes' here

L57: add a comma after "relationships"

Introduction

L76: delete "event"

L82: replace "Currently," with "Reportedly,"- these references are not 100% current. Also, replace "stranding" with "that strand".

L88: insert "where" in front of "as of"

L89: replace "incidence" with "prevalence"

L95: replace "seems to" with "may"

L97: replace "the specific" with "postulated"

L109: insert a comma after "internal"

Results

L308: insert "with" in front of "what was seen"

Discussion

L353: delete the comma after "ChHV5"

L371: insert a comma after "tumors"

L378-379: delete "to" after "targeted", insert "to" after "simultaneously"

Materials & Methods

L424: replace "are" with "were"; delete "are"

L523: insert "were" in front of "generated"

L525: replace "was" with "were" (the word "data" is plural)

> Table 2 is great, adds a lot to the paper!

> Figure 4 caption- why is "cytoskeletons" red?

Reviewer #2 (Remarks to the Author):

I am happy that all of the questions that I have raised have been adequately addressed

Response to Reviewers

We would like to again thank the reviewers for their careful consideration of the manuscript and for their thoughtful and constructive comments. We have implemented all requested changes, which we believe have helped to further improve the manuscript.

Author responses in **red text**.

REVIEWERS' COMMENTS:

Reviewer #1 (Remarks to the Author):

The manuscript is much improved by the revisions. A few more suggested revisions are below:

Abstract

L42: change to "immunohistochemical" **Done**

L53: define 'poor outcomes' here **Done**

L57: add a comma after "relationships" **Done**

Introduction

L76: delete "event" **Done**

L82: replace "Currently," with "Reportedly," - these references are not 100% current. Also, replace "stranding" with "that strand". **Done**

L88: insert "where" in front of "as of" **Not done, 'In contrast' already at start of sentence negating the need for this additional 'where'.**

L89: replace "incidence" with "prevalence" **Done**

L95: replace "seems to" with "may" **Done**

L97: replace "the specific" with "postulated" **Done**

L109: insert a comma after "internal" **Done**

Results

L308: insert "with" in front of "what was seen" **Done**

Discussion

L353: delete the comma after "ChHV5" **Done**

L371: insert a comma after "tumors" **Done**

L378-379: delete "to" after "targeted", insert "to" after "simultaneously" **Done**

Materials & Methods

L424: replace "are" with "were"; delete "are" **Done**

L523: insert "were" in front of "generated" **Done**

L525: replace "was" with "were" (the word "data" is plural) **Done**

> Table 2 is great, adds a lot to the paper! **Thank you**

> Figure 4 caption- why is "cytoskeletons" red? **Corrected**

Reviewer #2 (Remarks to the Author):

I am happy that all of the questions that I have raised have been adequately addressed **Thank you**